# NOTCH-mediated non-cell autonomous regulation of chromatin structure during senescence

Aled J. Parry [1], Matthew Hoare[1,2], Dóra Bihary[3], Robert Hänsel-Hertsch[1], Stephen Smith [4], Kosuke Tomimatsu[1], Elizabeth Mannion[1], Amy Smith[1], Paula D'Santos[1], I. Alasdair Russell[1], Shankar Balasubramanian [1,5], Hiroshi Kimura [6], Shamith A. Samarajiwa[3] & Masashi Narita [1]

Senescent cells interact with the surrounding microenvironment achieving diverse functional outcomes. We have recently identified that NOTCH1 can drive 'lateral induction' of a unique senescence phenotype in adjacent cells by specifically upregulating the NOTCH ligand JAG1. Here we show that NOTCH signalling can modulate chromatin structure autonomously and non-autonomously. In addition to senescence-associated heterochromatic foci (SAHF), oncogenic RAS-induced senescent (RIS) cells exhibit a massive increase in chromatin accessibility. NOTCH signalling suppresses SAHF and increased chromatin accessibility in this context. Strikingly, NOTCH-induced senescent cells, or cancer cells with high JAG1 expression, drive similar chromatin architectural changes in adjacent cells through cell–cell contact. Mechanistically, we show that NOTCH signalling represses the chromatin architectural protein HMGA1, an association found in multiple human cancers. Thus, HMGA1 is involved not only in SAHFs but also in RIS-driven chromatin accessibility. In conclusion, this study identifies that the JAG1–NOTCH–HMGA1 axis mediates the juxtacrine regulation of chromatin architecture.

[1] Cancer Research UK Cambridge Institute, University of Cambridge, Robinson Way, Cambridge CB2 0RE, UK. [2] Department of Medicine, Addenbrooke's Hospital, University of Cambridge, Cambridge CB2 0QQ, UK. [3] MRC Cancer Unit, Hutchison/MRC Research Centre, University of Cambridge, Cambridge Biomedical Campus, Cambridge CB2 0XZ, UK. [4] Department of Pathology, Addenbrooke's Hospital, University of Cambridge, Cambridge CB2 0QQ, UK. [5] Department of Chemistry, University of Cambridge, Lensfield Road, Cambridge CB2 1EW, UK. [6] Cell Biology Centre, Institute of Innovative Research, Tokyo Institute of Technology, Yokohama 226-8503, Japan. Correspondence and requests for materials should be addressed to S.A.S. (email: SS861@MRC-CU.cam.ac.uk) or to M.N. (email: masashi.narita@cruk.cam.ac.uk)

Cellular senescence is an autonomous tumour-suppressor mechanism that can be triggered by pathophysiological stimuli including replicative exhaustion, exposure to chemotherapeutic drugs and hyper-activation of oncogenes, such as RAS[1]. Persistent cell cycle arrest is accompanied by diverse transcriptional, biochemical and morphological alterations. These senescence hallmarks include increased expression and secretion of soluble factors (senescence-associated secretory phenotype (SASP))[2,3] and dramatic alterations to chromatin structure[1,4,5]. Importantly, the combination, quantity and quality of these features can vary depending on the type of senescence. Senescent cells have profound non-cell autonomous functionality. The SASP can have either protumorigenic or antitumorigenic effects and act in an autocrine or paracrine fashion[2,6–8]. In addition, we have recently identified that NOTCH signalling can drive a cell-contact-dependent juxtacrine senescence[9].

The NOTCH signalling pathway is involved in a wide array of developmental and (patho-)physiological processes. NOTCH has roles in differentiation and stem cell fate[10] and perturbations have

been linked to tumorigenesis where NOTCH can have either oncogenic or tumour-suppressive functionality[11]. The pathway involves proteolytic cleavage of the NOTCH receptor upon contact-mediated activation by a ligand of the JAGGED (JAG) or DELTA family on the surface of an adjacent cell. The cleaved NOTCH-intracellular domain translocates to the nucleus where, together with transcriptional co-activators such as mastermind-like 1 (MAML1), it drives transcription of canonical target genes, including the HES and HEY family of transcription factors[10]. NOTCH signalling has also been shown to induce a type of senescence, NOTCH-induced senescence (NIS), where cells are characterised by distinct SASP components[9,12]. Recently, we showed that during NIS there is a dramatic and specific upregulation of JAG1 that can activate NOTCH1 signalling and drive NIS in adjacent cells ('lateral induction')[9].

During senescence, particularly in oncogenic RAS-induced senescent (RIS) fibroblasts, characteristic changes to chromatin culminate in the formation of senescence-associated heterochromatic foci (SAHFs)[13], layered structures facilitated by spatial

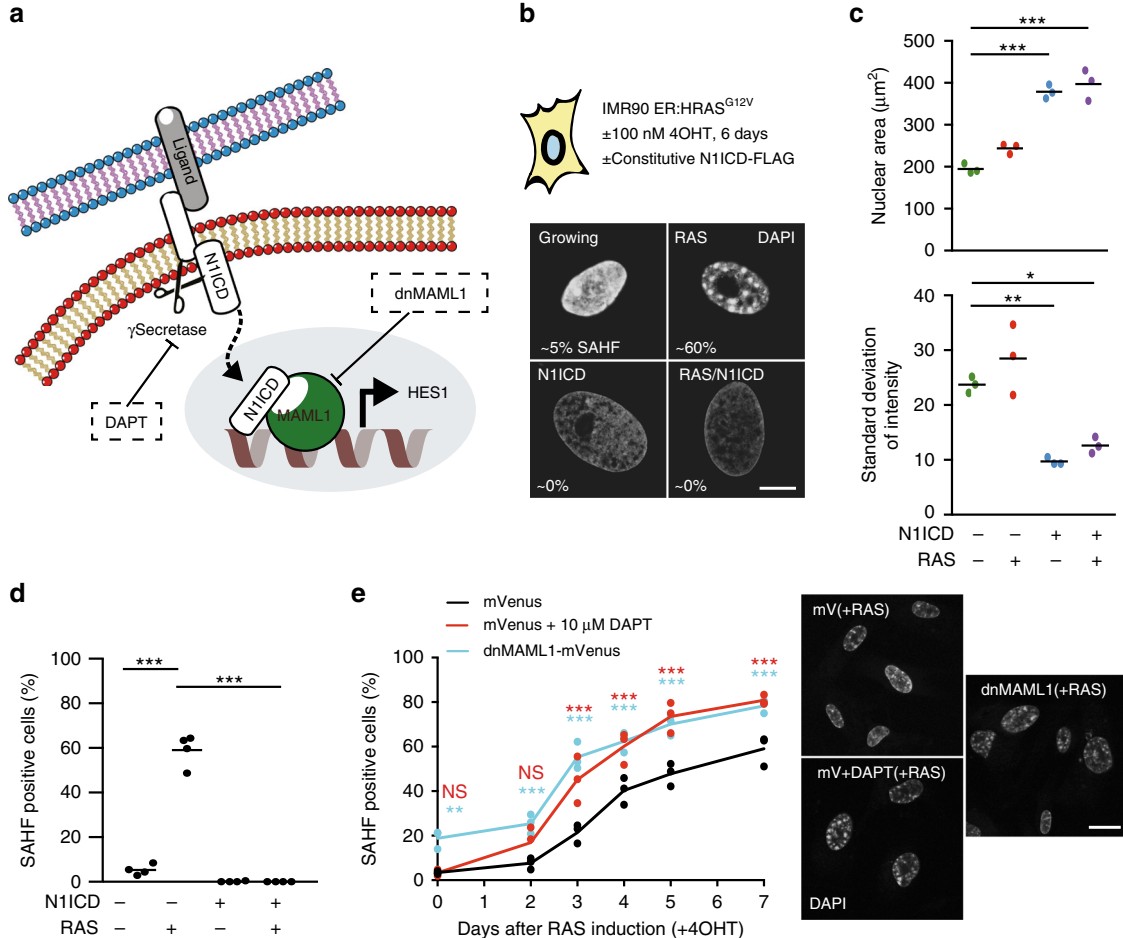

**Fig. 1** NOTCH1 signalling has a chromatin 'smoothening' effect that blocks SAHF. **a** Diagram illustrating the NOTCH1 signalling pathway, which can be repressed chemically using DAPT or genetically by expressing dominant-negative MAML1 (dnMAML1). **b** IMR90 ER:HRAS^G12V cells were infected with control vector or N1ICD-FLAG and incubated with ±100 nM 4OHT for 6 days. Representative images of nuclei stained with DAPI for the conditions indicated (scale bar = 10 μm). Percentage indicates the number of SAHF-positive cells within the population (see **d**). **c, d** Quantification of nuclear area, standard deviation of DAPI intensity (**c**) and the number of SAHF-positive cells (**d**) for the conditions indicated in **b**. Lines indicate the mean value of each replicate. n = 3 (**c**) and n = 4 (**d**) biologically independent replicates. Values of individual replicates for nuclear area and standard deviation are shown in Supplementary Fig. 1b, d. **e** Time series analysis of SAHF-positive nuclei following the addition of 100 nM 4OHT to IMR90 ER:HRAS^G12V cells in the presence or absence of ectopic dnMAML1 or 10 μM DAPT (left). n = 3 biologically independent replicates. Representative DAPI images of the indicated conditions (+RAS = 7 days of 4OHT treatment; scale bar = 25 μm). **c–e** Statistical significance calculated using one-way ANOVA with Tukey's correction for multiple comparisons. *p ≤ 0.05, **p ≤ 0.01, ***p ≤ 0.001

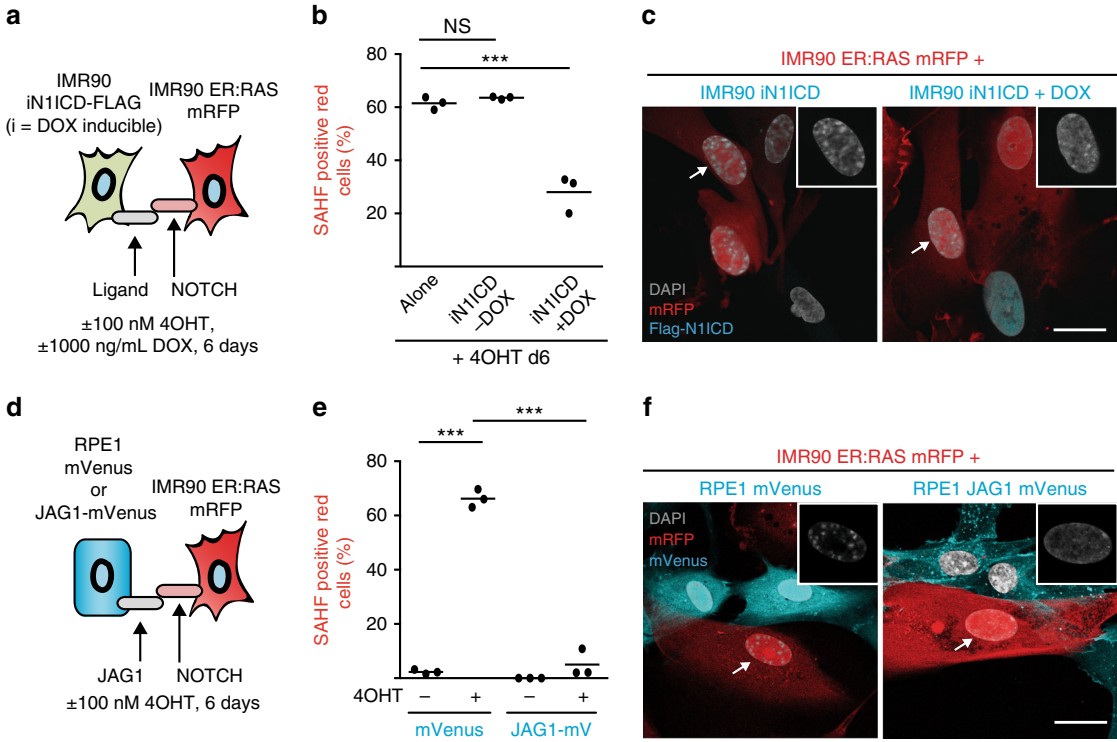

**Fig. 2** NOTCH1 and JAG1 can non-autonomously repress SAHF formation in adjacent cells. **a** Schematic showing experimental set-up. IMR90 cells expressing doxycycline (DOX)-inducible N1ICD-FLAG were cultured with IMR90 ER:HRAS[G12V] cells expressing mRFP with 100 nM 4OHT ± 1000 ng/mL DOX for 6 days. **b** Quantification of SAHF-positive red cells for the experiment outlined in **a**. Alone: mono-cultured IMR90 ER:HRAS[G12V] cells; iN1ICD: DOX-inducible N1ICD-FLAG. **c** Representative images of co-cultures indicated (scale bar = 25 μm). Insets are unmerged DAPI images of the indicated cells (arrows). **d** Schematic showing experimental set-up. IMR90 ER:HRAS[G12V] cells expressing mRFP were co-cultured with RPE1 cells stably expressing either mVenus or JAG1-mVenus for 6 days ±100 nM 4OHT. **e** Quantification of SAHF-positive red cells for the experiment outlined in **d**. **f** Representative images of co-cultures indicated (scale bar = 25 μm). Insets are unmerged DAPI images of the indicated cells (arrows). Note DAPI foci in RPE1 cells are not SAHFs. **b**, **e** Lines indicate the mean value of individual replicates. n = 3 biologically independent replicates for all conditions. Statistical significance calculated using one-way ANOVA with Tukey's correction for multiple comparisons; ***p ≤ 0.001, NS = not significant

rearrangement of existing heterochromatin[14]. Other alterations include the formation of senescence-associated distension of satellites (SADS)[15].

SAHF formation is dependent on chromatin-bound high-mobility group A (HMGA) proteins, particularly HMGA1[16]. These are a family of architectural proteins, consisting of HMGA1 and HMGA2, which bind to the minor groove of AT-rich DNA via three AT-hook domains to alter chromatin structure[17,18]. Despite a critical role in the formation of SAHFs during senescence, HMGA proteins are also important during development where they promote tissue growth[19,20] and regulate differentiation[21–24]. Furthermore, many studies have demonstrated an association between high HMGA1 expression and aggressive tumour biology[25,26].

Chromatin accessibility at regulatory elements including promoters and enhancers is highly correlated with biological activity[27]. High-throughput sequencing using FAIRE-seq, a method that identifies open and closed chromatin based on phenol separation[28], has revealed that, in cells that have undergone replicative senescence, previously heterochromatic domains enriched for various repeat elements become more accessible while euchromatic domains undergo condensation[29]. However, it remains unknown how chromatin accessibility is altered in RIS and NIS cells.

Here we characterise the chromatin phenotype in RIS and NIS cells. We demonstrate that these two types of senescent cells exhibit distinct chromatin structures at microscopic and nucleosome scales.

Both gain multiple chromatin accessible regions, which are often exclusive between RIS and NIS. Strikingly, we find that autonomous and non-cell autonomous activation of the NOTCH signalling pathway in RIS cells can repress SAHFs and the formation of RIS-driven chromatin-accessible regions, partially by transcriptional repression of HMGA1. Our study demonstrates that chromatin structure and the nucleosome landscape can be regulated through juxtacrine signalling. The relationship between these two prominent tumour-associated genes, HMGA1 and NOTCH1, may also have prognostic value in vivo.

## Results

**NOTCH1 reprogrammes chromatin structure and abrogates SAHFs.** We have previously demonstrated that ectopic NOTCH1-intracellular domain (N1ICD), an active form of NOTCH1 (Fig. 1a), can drive NIS that is distinct from RIS in terms of SASP composition[9] and noticed that NIS cells also have a unique chromatin structure.

To examine the relationship between NOTCH1 and chromatin structure, we introduced ectopic N1ICD into IMR90 human diploid fibroblasts (HDFs) stably expressing a 4-hydroxytamoxifen (4OHT)-inducible oestrogen receptor–oncogenic HRAS fusion protein (IMR90 ER:HRAS[G12V] cells)[30]. Ectopic expression of N1ICD alone induced senescence with dramatically enlarged nuclei, even larger than in RIS (Fig. 1b, c). Similarly to RIS, NIS cells formed SADS, a more common chromatin feature of

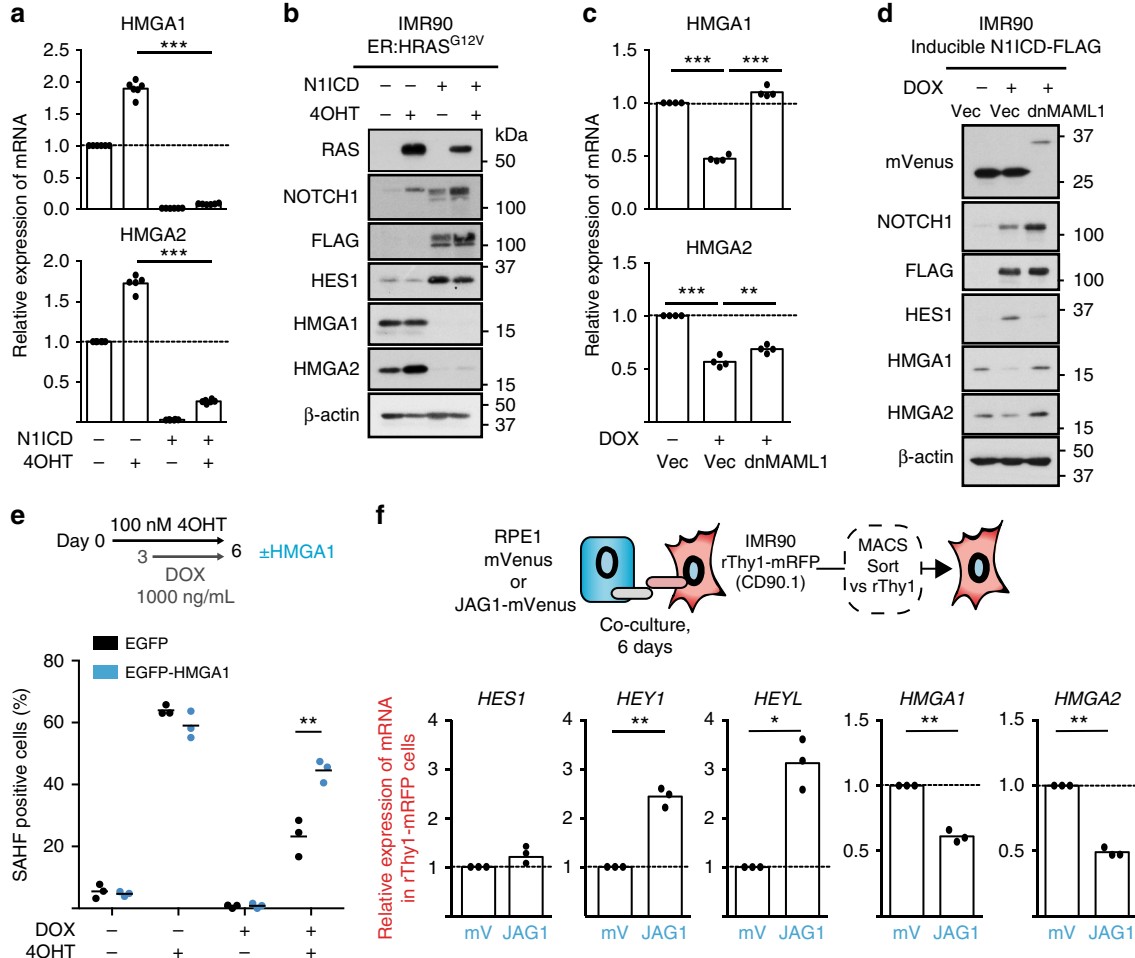

**Fig. 3** NOTCH1 signalling represses SAHF formation partially by repressing HMGA proteins. **a, b** qRT-PCR ($n = 6$) (**a**) and immunoblotting (**b**) for the indicated mRNA and proteins in IMR90 ER:HRAS$^{G12V}$ cells stably infected with control vector or N1ICD-FLAG ± 100 nM 4OHT for 6 days. **c, d** qRT-PCR ($n = 4$) (**c**) and immunoblotting (**d**) of IMR90 cells expressing doxycycline (DOX)-inducible N1ICD-FLAG (iN1ICD) and infected with a mVenus control vector or dnMAML1-mVenus ± 1000 ng/mL DOX for 3 days. **e** Quantification of SAHFs in IMR90 cells expressing ER:HRAS$^{G12V}$, iN1ICD and EGFP or an EGFP-HMGA1 fusion ± 100 nM 4OHT for 6 days and ±1000 ng/mL DOX for 3 days. **f** qRT-PCR for the indicated mRNA in IMR90 cells expressing rThy1-mRFP and co-cultured with RPE1 cells expressing mVenus or JAG1-mVenus, isolated using MACS ($n = 3$). Statistical significance calculated using one-way ANOVA with Tukey's correction for multiple comparisons (**a, c**) or two-sample $t$-test (**e, f**). *$p \leq 0.05$, **$p \leq 0.01$, ***$p \leq 0.001$

senescence than SAHFs (Supplementary Fig. 1a)[15]. However, in marked contrast to RIS, NIS cells lacked SAHFs (Fig. 1b, d).

To ask whether NIS cells simply lack SAHFs or whether N1ICD actively modulates chromatin structure, we expressed N1ICD in the presence of HRAS$^{G12V}$ induced using 100 nM of 4OHT for 6 days. Interestingly, N1ICD in the context of RIS also resulted in a dramatic enlargement of nuclei but a complete ablation of SAHF formation (Fig. 1b–d). This was emphasised by a 'smoothening' of chromatin as indicated by a marked reduction in the standard deviation of 4,6-diamidino-2-phenylindole (DAPI) signal measured within individual nuclei (Fig. 1b, c; Supplementary Fig. 1b-d), We have previously shown that ectopic N1ICD in the RIS context results in senescence with SASP composition broadly similar to NIS[9]. Thus our data indicate that NOTCH is dominant over RIS in terms of chromatin phenotype as well as SASP composition.

In IMR90 ER:HRAS$^{G12V}$ cells, RIS develops progressively over a time period of ~6 days following the addition of 4OHT[30]. NOTCH1 signalling is temporally regulated during RIS, where cleaved and active N1ICD is transiently upregulated before downregulation at full senescence[9]. To examine the temporal effects of NOTCH1 signalling on SAHF formation, we performed

a time course experiment in IMR90 ER:HRAS$^{G12V}$ cells. Cells were retrovirally infected with a dominant-negative form of MAML1 fused to mVenus (dnMAML1-mVenus) or treated with the γ-secretase inhibitor *N*-[(3,5-difluorophenyl)acetyl]-L-alanyl-2-phenyl]glycine-1,1-dimethylethyl ester (DAPT) to repress downstream signalling by N1ICD (Fig. 1a). We found that a greater number of SAHF-positive cells were formed and that these accumulated at earlier time points when NOTCH1 signalling was repressed (Fig. 1e). Furthermore, a dose-dependent effect was evident where higher concentrations of DAPT resulted in a greater proportion of cells developing SAHF during RIS (Supplementary Fig. 1e). SAHFs are not typically prominent in DNA damage-induced senescence (DDIS) in IMR90 cells[1]. However, DAPT significantly promoted SAHF formation in DDIS (by etoposide) (Supplementary Fig. 1f). To determine whether NOTCH1 activity can reverse SAHF after they have formed, we infected IMR90 cells with doxycycline (DOX)-inducible N1ICD-FLAG and constitutive HRAS$^{G12V}$. The addition of DOX after the establishment of senescence was sufficient to reduce the number of SAHF-positive cells and the standard deviation of DAPI signal, suggesting some degree of reversibility (Supplementary Fig. 1g). Together, our data suggest that NOTCH

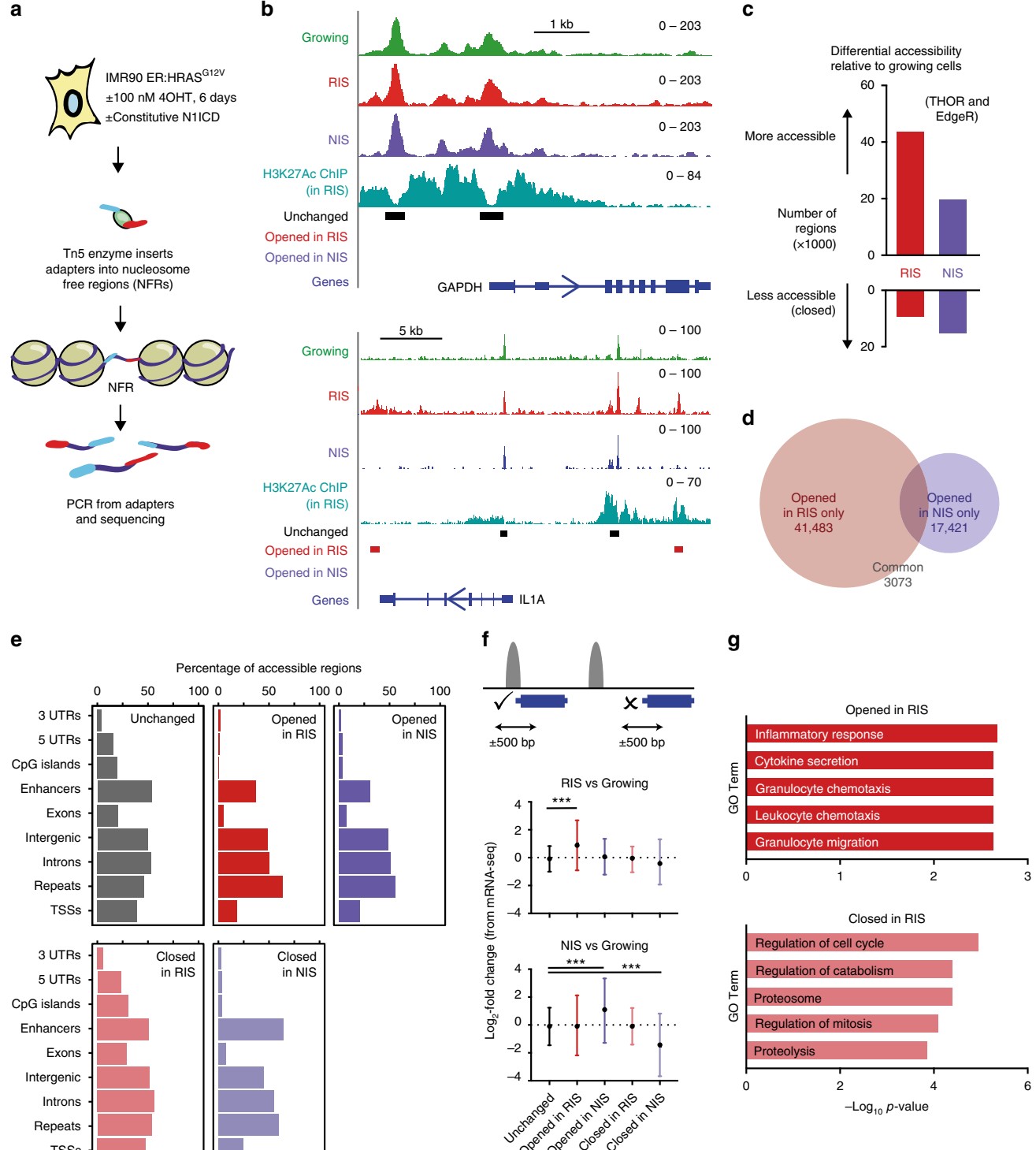

**Fig. 4** Chromatin accessibility reflects gene transcription in RIS and NIS cells. **a** Diagram illustrating the method of ATAC-seq. **b** Genome browser images showing normalised ATAC-seq coverage and an active enhancer-associated histone modification (H3K27ac) in the cell conditions indicated around the *GAPDH* and *IL1A* genes. Unchanged = accessibility unaltered in RIS or NIS cells relative to growing; Opened in RIS = more accessible in RIS vs. growing cells; Opened in NIS = more accessible in NIS vs. growing cells (by edgeR and THOR). **c** Number of regions that become more accessible and less accessible in RIS and NIS cells relative to growing cells (intersect of edgeR and THOR). **d** Number of regions that are more accessible in RIS that are overlapping with a region that is more accessible in NIS (vs. growing). **e** Annotation of more (opened) and less (closed) accessible regions in RIS and NIS cells to genomic regions. **f** Accessible regions within the indicated subsets were annotated to genes if within 500 bp of a TSS. The average log$_2$-fold expression change (by mRNA-seq) of genes in RIS or NIS cells relative to growing cells is plotted. Values are mean ± s.d. Statistical significance calculated using a one-way ANOVA with Tukey's correction for multiple comparisons. ***$p \leq 0.001$. **g** Gene ontology analysis (GO Biological Process 2015) using the TSS proximal accessible regions described in **f**

signalling has a chromatin 'smoothening' effect that antagonises SAHF formation.

**Non-cell autonomous regulation of SAHFs**. N1ICD-expressing cells can induce NIS in adjacent normal cells, at least in the case of IMR90 fibroblasts[9]. To determine whether N1ICD-expressing cells can also alter chromatin structure in adjacent cells, we performed co-cultures between mRFP1-expressing IMR90 ER: HRAS[G12V] and IMR90 cells expressing DOX-inducible N1ICD-FLAG in the presence and absence of 4OHT and DOX (Fig. 2a). Strikingly, co-culture with N1ICD-expressing IMR90 cells was sufficient to repress SAHF formation in adjacent RIS (red) cells (Fig. 2b, c).

Of the canonical NOTCH1 ligands, we have previously observed a strong and unique upregulation of JAG1 following ectopic N1ICD expression, which we found to be responsible for the juxtacrine transmission of NIS[9]. We reasoned that N1ICD-mediated upregulation of JAG1 and subsequent 'lateral induction' of NOTCH1 signalling is a likely mechanism by which SAHFs are regulated non-autonomously. To test this hypothesis, we expressed ectopic JAG1 fused to mVenus (JAG1-mVenus) in retinal pigment epithelial (RPE1) cells. We confirmed cell surface expression of ectopic JAG1 by flow cytometry (Supplementary Fig. 2a) before co-culturing with mRFP1-expressing IMR90 ER: HRAS[G12V] cells. RPE1 JAG1-mVenus cells, but not control RPE1 cells, significantly repressed the formation of SAHFs (Fig. 2e, f). Note that this repression did not occur when these two types of cells were co-cultured without physical contact in a transwell format (Supplementary Fig. 2b). Our data suggest a mechanism by which lateral induction of NOTCH signalling by JAG1 can block SAHFs in the context of RIS; i.e. higher-order chromatin structure can be regulated through cell–cell contact.

**NOTCH signalling represses the expression of *HMGA* genes.** To unravel the mechanisms underpinning NOTCH1-dependent repression of SAHFs, we re-analysed previously published RNA-seq data generated from IMR90 cells expressing HRAS[G12V] and N1ICD[9]. We found that N1ICD dramatically represses the expression of *HMGA1* and *HMGA2* (Supplementary Fig. 3a), critical components of SAHF structure[16].

To validate that NOTCH1 signalling represses HMGAs, we introduced constitutive N1ICD into IMR90 ER:HRAS[G12V] cells. Ectopic N1ICD significantly repressed HMGA1 and HMGA2 at an mRNA and protein level in both the presence and absence of 4OHT-induced HRAS[G12V] (Fig. 3a, b). The enforced expression of N1ICD after senescence establishment also resulted in the reduction of HMGA1 albeit to a lesser extent than pre-senescence N1ICD expression (Supplementary Fig. 1g). N1ICD has a similar effect on HMGA1 and 2 protein levels when expressed in other cell lines in the absence of HRAS[G12V], suggesting a conserved mechanism (Supplementary Fig. 3b). In the DOX-inducible N1ICD-FLAG system, inhibition of NOTCH1 signalling by co-expression of dnMAML1-mVenus was sufficient to rescue N1ICD-mediated repression of HMGA1 and HMGA2 (Fig. 3c, d), suggesting the effect is dependent on the canonical pathway of NOTCH signalling.

Finally, we used IMR90 ER:HRAS[G12V] cells expressing DOX-inducible N1ICD-FLAG to investigate whether ectopic re-expression of EGFP-tagged HMGA1 is sufficient to rescue SAHFs. The introduction of EGFP-HMGA1 resulted in a partial, but significant, rescue of SAHF-positive cells when cells were treated with DOX and 4OHT (Fig. 3e).

Collectively, our data suggest that NOTCH1 signalling represses the formation of SAHFs at least partially by inhibiting HMGAs.

**Non-cell autonomous inhibition of HMGAs**. To determine whether HMGAs are repressed non-autonomously by JAG1 expressing cells, we performed further co-cultures between RPE1 cells retrovirally infected with JAG1-mVenus and IMR90 cells ectopically expressing a cell surface marker, rat-Thy1, allowing for subsequent isolation using magnetic-activated cell sorting (MACS) (Fig. 3f). As expected, IMR90 cells co-cultured with JAG1-expressing cells upregulated canonical NOTCH1 target genes, *HEY1* and *HEYL*. Both *HMGA1* and *HMGA2* were significantly repressed in the same IMR90 cells (Fig. 3f), demonstrating that HMGA proteins can be repressed non-cell autonomously.

**Altered chromatin accessibility in RIS and NIS**. To investigate whether NOTCH1 influences chromatin structure at a higher resolution, we employed ATAC-seq (assay for transposase-accessible chromatin using sequencing)[31]. This method exploits a hyperactive Tn5 transposase that inserts sequencing adapters into regions of accessible chromatin. Following adapter-primed PCR amplification, these regions were sequenced to identify accessible regions of chromatin genome wide (Fig. 4a).

We generated at least three replicates from IMR90 ER: HRAS[G12V] cells expressing N1ICD-FLAG or a control vector and induced with 4OHT or not. For simplicity, these conditions were labelled as 'Growing', 'RIS', 'NIS' and 'N+RIS' (expressing both N1ICD and RAS). Using a previously published normalisation approach[32], we generated normalised coverage files that appeared comparable to each other, especially around house-keeping genes (Fig. 4b). Most of the samples, excluding a single replicate from the NIS and N+RIS conditions (which were excluded from downstream analysis), were of high quality with a 'reads in peaks' percentage (RiP%) of >10% (Supplementary Fig. 4a). Replicates clustered well by unbiased principal component analysis (PCA) (Supplementary Fig. 4b). Moreover, our samples clustered with publically available ATAC-seq and DNase-seq data generated from IMR90 cells (Supplementary Fig. 4c), but separated from other cell types (BJ, HaCaT, MCF710A and HEKn cells).

Using MACS peak calling, we found that the number of peaks identified in each replicate of a condition was similar and that, in general, chromatin accessibility was dramatically increased in RIS (145,649 consensus peaks detected in ≥2 replicates) and NIS cells (149,877 peaks) relative to growing cells (83,920 peaks) (Supplementary Fig. 5a). To quantitatively identify regions of altered accessibility in RIS and NIS cells relative to growing cells, we performed differential binding analysis using both edgeR[33,34] and THOR[35] before taking only regions identified by both methods for downstream analysis (Supplementary Data 1). Using this stringent approach, we identified 44,556 regions that become significantly more accessible (opened) and 9603 regions that become significantly less accessible (closed) in RIS cells relative to growing cells. In NIS cells, 20,499 regions became more accessible and 15,444 regions less accessible (Fig. 4c). Despite the robust gain of chromatin accessibility in both types of senescence, there were relatively few shared sites (Fig. 4d).

A previous study mapping chromatin accessibility in replicatively senescent cells using FAIRE-Seq found that gene-distal regions, especially repeat regions, become relatively more open whereas genic regions become closed compared to growing fibroblasts[29]. Consistently, regions of increased accessibility in RIS and NIS cells were enriched at gene-distal sites

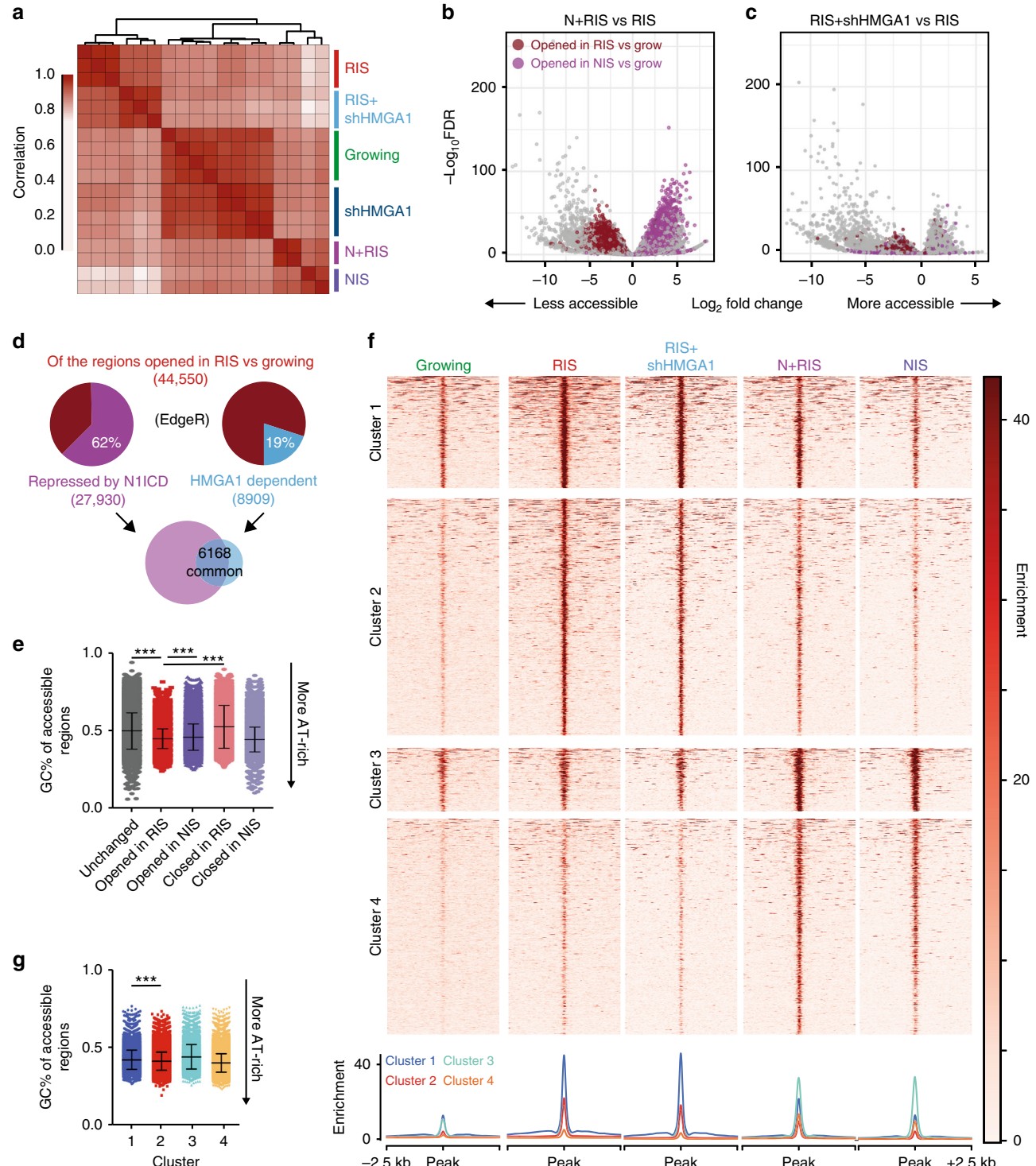

**Fig. 5** Ectopic N1ICD and HMGA1 knockdown antagonise chromatin opening in RIS. **a** Unbiased clustering of replicates for the conditions indicated. **b**, **c** Volcano plots showing regions of altered accessibility in N+RIS cells (**b**) and RIS+shHMGA1 (**c**) cells relative to RIS cells. Regions that are also opened in RIS (red) and NIS (purple) relative to growing cells are indicated. **d** Number of novel accessible regions in RIS (identified in Fig. 4c) that are repressed by N1ICD (significantly reduced in N+RIS vs. RIS) and are HMGA1 dependent (significantly reduced in RIS+shHMGA1 vs. RIS). The Venn diagram shows the number of regions repressed by N1ICD that are overlapped by a region dependent on HMGA1. **e** Genomic GC percentage of the accessible regions indicated. **f** K-means clustered heatmap showing the enrichment of normalised reads around accessible regions altered in any of the conditions relative to growing cells. **g** Genomic GC percentage of the clusters indicated, identified in **f**. **e**, **g** Mean ± s.d is plotted. Statistical significance calculated using one-way ANOVA with Tukey's correction for multiple comparisons; ***$p \le 0.001$

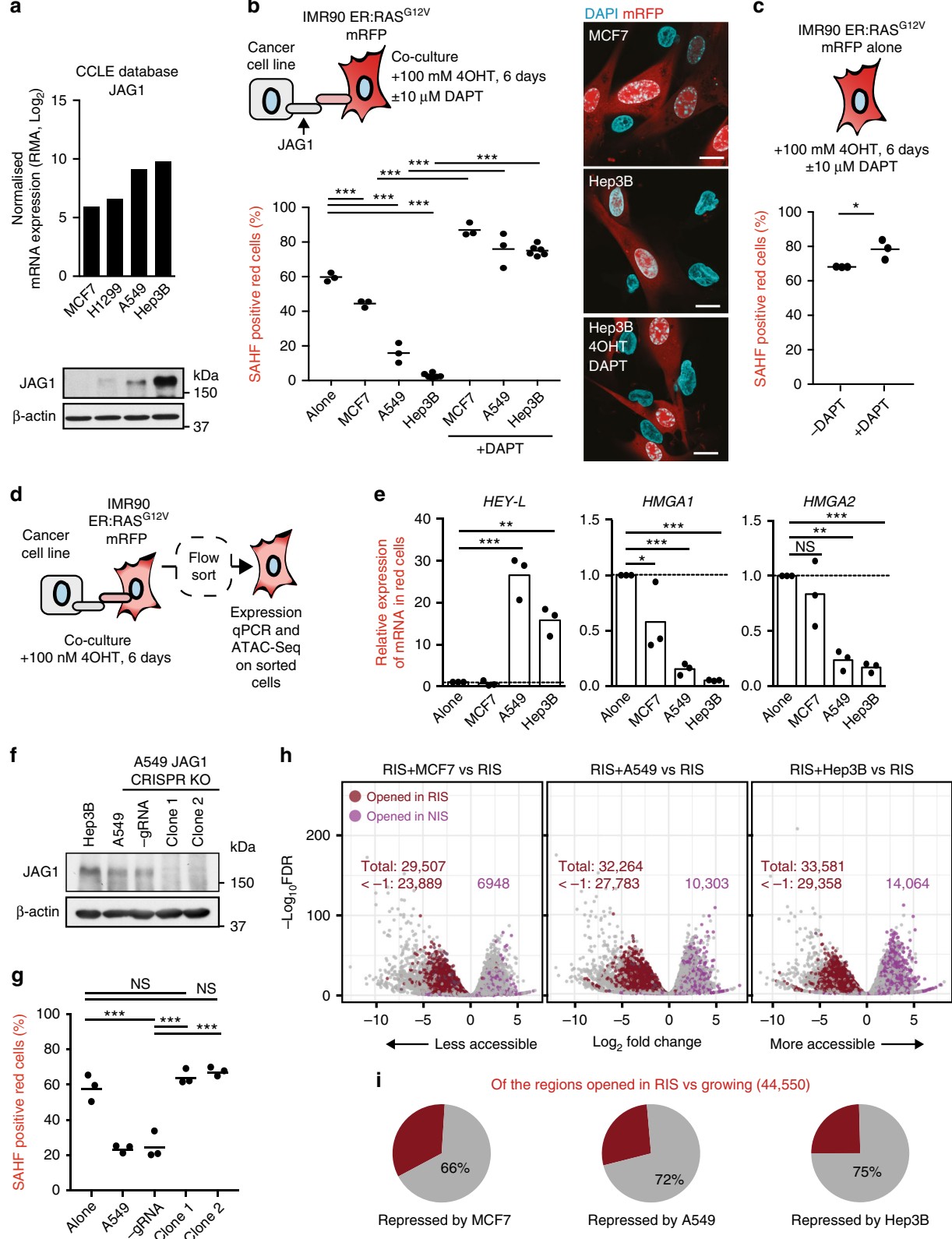

(Supplementary Fig. 5b) with the majority of opened regions mapping to enhancer, intergenic, intronic and repeat regions (Fig. 4e). Many of these repeat regions were further annotated as long interspersed elements, long-terminal repeats, short interspersed elements and simple repeat regions (Supplementary Fig. 5c, d), although these values may be underestimated due to the exclusion of multi-mapping reads from our data. Many of the regions that became less accessible in RIS cells relative to growing cells were closer to transcriptional start sites (TSSs) (Supplementary Fig. 5b) and mapped to exons, CpG-islands and untranslated regions (UTRs) (Fig. 4e). In contrast to replicative senescence[29], regions of decreased accessibility in NIS cells mostly mapped to

**Fig. 6** Tumour cells can repress SAHFs and chromatin opening in adjacent RIS fibroblasts. **a** Normalised mRNA expression values from the Cancer Cell Line Encyclopedia (CCLE) and immunoblotting of JAG1 in the tumour cell lines indicated. **b** Quantification of SAHFs in IMR90 ER:HRAS$^{G12V}$ cells expressing mRFP co-cultured with the tumour cell lines indicated for 6 days +100 nM 4OHT ± 10 μM DAPT (left) and representative images (right) (scale bar = 25 μm); n = 3 biological replicates except for Hep3B cultures where n = 6. **c** SAHF quantification in IMR90 ER:HRAS$^{G12V}$ cells expressing mRFP+100 nM 4OHT±10 μM DAPT. **d** Schematic showing experimental set-up. IMR90 ER:HRAS$^{G12V}$ cells expressing mRFP were cultured with tumour cell lines +100 nM 4OHT before flow sorting to isolate red cells. **e** qRT-PCR of mRNA isolated from the cells described in **c**. n = 3 biological replicates. **f** Immunoblotting of JAG1 in the cells indicated. **g** Quantification of SAHFs in IMR90 ER:HRAS$^{G12V}$ expressing mRFP co-cultured with the tumour cells indicated for 6 days +100 nM 4OHT. n = 3 biological replicates. **h** Volcano plots showing regions of altered accessibility in RIS cells co-cultured with MCF7, A549 and Hep3B cells (as in **c**) relative to RIS cells cultured alone. Regions that also become more accessible in RIS (red) and NIS (purple) vs. growing are indicated where numbers indicate the total number of significant alterations (log$_2$ fold change <−0.58 or >0.58 and FDR < 0.01) and <−1 indicates the number with a log$_2$ fold change of <−1. **i** Number of more accessible regions in RIS (identified in Fig. 4c) that are repressed by co-culture with MCF7, A549 and Hep3B. **b, c, e, g** Statistical significance calculated using one-way ANOVA with Tukey's correction for multiple comparisons; *p ≤ 0.05, **p ≤ 0.01, ***p ≤ 0.001, NS = not significant

gene-distal elements (Supplementary Fig. 5b, Fig. 4e). Therefore, while RIS largely mirrors replicative senescence, NIS is characterised by remodelling (both opening and closing) of gene-distal regions.

**Altered accessibility of genes reflects expression**. Chromatin accessibility at regulatory elements has been correlated with gene expression[27]. To determine whether genic alterations to chromatin accessibility in RIS and NIS reflects gene expression, we assigned regions of altered accessibility (opened or closed in RIS or NIS) to genes if within 500 bp of a TSS (Fig. 4f). On average, genes that were opened in RIS relative to growing cells were also transcriptionally upregulated by mRNA-seq in RIS relative to growing cells (Fig. 4f, top). Genes that were opened in NIS cells were transcriptionally upregulated in NIS cells, while less accessible genes were transcriptionally repressed (Fig. 4f, bottom). Consistent with our previous RNA-seq data[9], genes that became more accessible in RIS were significantly enriched within gene ontology (GO) terms such as 'inflammatory response' and 'cytokine secretion', reflecting the inflammatory secretome produced by RIS cells (Fig. 4g). Genes that became less accessible in RIS were enriched within GO terms such as 'regulation of cell cycle' (Fig. 4g), perhaps reflecting non-proliferative features of RIS (although average gene expression of this gene set was not significantly altered). Unbiased motif enrichment analysis revealed that regions opened in RIS were highly enriched for the C/EBPβ-binding motif (Supplementary Fig. 5e), consistent with the important role of C/EBPβ in regulating the inflammatory SASP[3]. Regions opened in NIS were enriched with the RBP-J-binding motif (Supplementary Fig. 5 f), a critical DNA-binding factor downstream of NOTCH signalling[11]. Normalised ATAC-seq coverage files, when viewed using a genome browser (Fig. 4b, Supplementary Fig. 6a-d), demonstrated increased accessibility around transcriptionally activated genes. We also noted that, while the accessibility at many promoters was unaltered in RIS cells, some transcriptionally activated genes, such as IL1A and HMGA1, were proximal to enhancer elements that became more accessible (Fig. 4b, Supplementary Fig. 6a). Together, these data demonstrate that RIS and NIS cells have unique open chromatin landscapes and that (gene proximal) alterations reflect their transcriptional landscapes.

**NOTCH signalling antagonises chromatin opening in RIS**. By unbiased clustering of ATAC-seq data, we observed a greater correlation between NIS and N+RIS cells than between RIS and N+RIS cells (Fig. 5a). This suggests a dominant effect of N1ICD over RAS on the nucleosome scale, consistent with our previous observations for SASP components[9] and SAHFs (Fig. 1d). To determine whether NOTCH1 signalling can repress the chromatin alterations observed in RIS in favour of a 'NIS-like' chromatin landscape, we focussed on the 44,556 regions that

became significantly more accessible in RIS cells relative to growing cells (referred to as 'RIS-driven accessible regions', Fig. 4c) and the 20,499 regions that became significantly more accessible in NIS cells relative to growing cells (referred to as 'NIS-driven accessible regions', Fig. 4c). By comparing chromatin accessibility of N+RIS cells with RIS cells we found that formation of many RIS-driven accessible regions (62.7%) were repressed by N1ICD expression (Fig. 5b). N1ICD expression also increased the accessibility of NIS-driven accessible regions (Fig. 5b). When viewed in the genome browser, it was evident that N1ICD expression can repress the formation of accessible regions located at enhancer elements upstream of the HMGA1 promoter in RIS cells (Supplementary Fig. 6a), although we failed to detect any alterations at the HMGA2 locus (Supplementary Fig. 6b), providing a potential mechanism for NOTCH1-mediated repression of HMGA1.

HMGA proteins have previously been shown to affect chromatin compaction. To determine whether repression of HMGA1 is a mechanism by which N1ICD can repress formation of RIS-driven accessible regions, we generated additional ATAC-seq samples from IMR90 ER:HRAS$^{G12V}$ cells expressing a short hairpin against HMGA1[16] and treated with 4OHT, hereafter referred to as 'RIS+shHMGA1'. By comparing RIS+shHMGA1 with RIS, we identified 8909 RIS-driven accessible regions that were dependent on HMGA1 (Fig. 5c). Of these, 69.9% (6168) were also repressed by N1ICD (Fig. 5d). These analyses illustrate that a subset of RIS-driven accessible regions can be repressed by N1ICD, possibly by HMGA downregulation. However, HMGA1 knock-down was not sufficient to induce the formation of NIS-driven accessible regions (Fig. 5c), suggesting an HMGA1-independent mechanism in the formation of these sites. RIS-driven accessible regions (opened in RIS) were significantly more AT-rich than NIS-driven accessible regions (opened in NIS) or regions with reduced accessibility (Fig. 5e), supporting the involvement of HMGA1 in the formation of RIS-driven accessible regions.

To validate the above approach, we used our normalised coverage files to perform unbiased k-means clustering centred around accessible regions that were altered in either RIS or NIS cells (opened or closed relative to growing cells) (Fig. 5f). Accessible regions separated into clusters that were dominated by either the RIS (clusters 1 and 2) or NIS (clusters 3 and 4) conditions. Strikingly, the signal in RIS-dominated clusters, cluster 2 in particular, was reduced in the N+RIS and RIS+shHMGA1 conditions when compared to the RIS condition (Fig. 5f). Consistently, cluster 2 was more AT-rich than cluster 1 (Fig. 5g), supporting a role for HMGA1. Notably, while peaks in clusters 3 and 4 were increased in the N+RIS condition, they did not increase in the RIS+shHMGA1 condition, reinforcing an HMGA1-independent mechanism of chromatin opening in NIS (Fig. 5f). Therefore, in line with microscopic SAHF structures, N1ICD alters chromatin structure in RIS at the nucleosome scale in part by repressing HMGA1 expression.

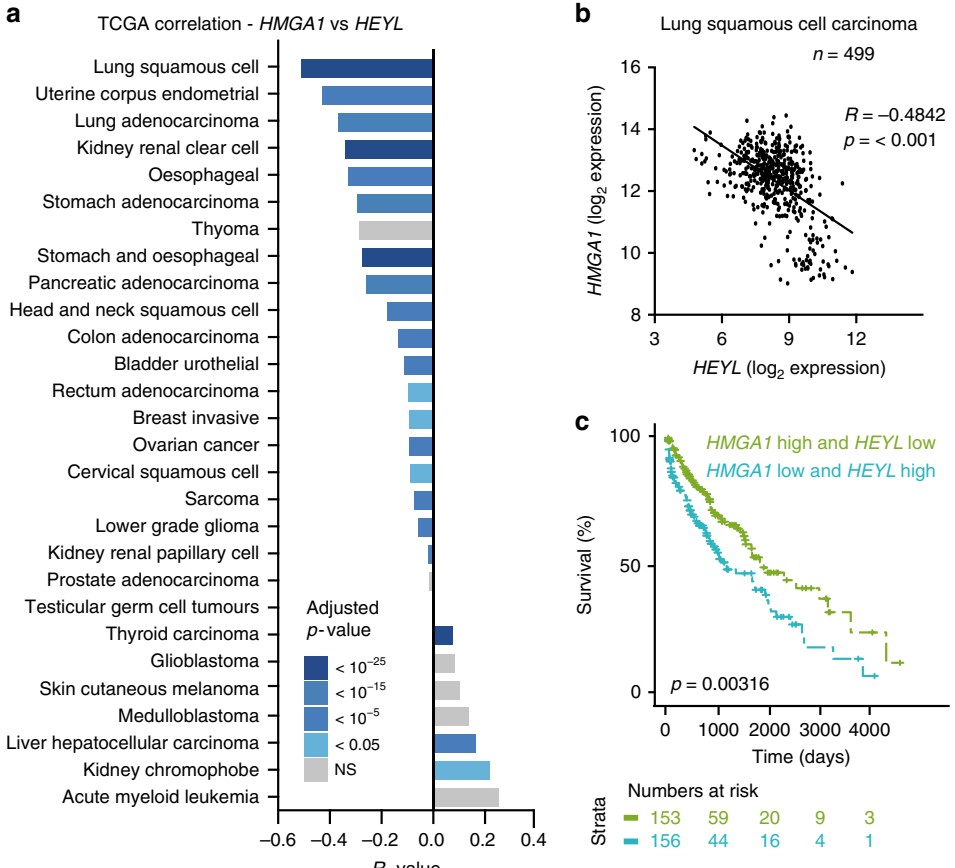

**Fig. 7** HEYL and HMGA1 gene expression values anti-correlate in multiple human tumour types. **a** Pan-cancer analysis of the TCGA database. Correlation between *HMGA1* and *HEYL* is plotted in the indicated tumour types. Colours represent Bonferroni adjusted *p*-values on Pearson's correlation *p*-values. NS = not significant. **b** Log₂ expression of *HMGA1* against *HEYL* in lung squamous cell carcinoma (LSCC). **c** Kaplan–Meier plot showing survival of LSCC patients stratified by *HMGA1* and *HEYL* gene expression

**Non-cell autonomous regulation of SAHFs by tumour cells.** Both HMGA1 and NOTCH1 can act as oncogenes or tumour suppressors in a context-dependent manner. We reasoned that the relationship between these two genes might also be important in the tumour microenvironment and asked whether tumour cells expressing JAG1 can affect HMGA1 expression and chromatin structure in adjacent fibroblasts.

To answer this question, we used the Cancer Cell Line Encyclopedia[36] to identify tumour cell lines that express low (MCF7), medium (A549) and high (Hep3B) levels of JAG1, which we confirmed by immunoblotting (Fig. 6a). Co-culture of tumour cell lines with IMR90 cells expressing both ER:HRAS^G12V and mRFP1 in the presence of 4OHT was sufficient to repress SAHF formation in red (RIS) cells in a contact-dependent manner (Fig. 6b; Supplementary Fig. 7a). The number of SAHF-positive red cells inversely correlated with the level of JAG1 expressed by the tumour cell lines (Fig. 6b). Non-autonomous inhibition of SAHF formation in the co-culture system was completely abrogated by DAPT, suggesting the effect is dependent on the canonical NOTCH pathway (Fig. 6b, c). Consistent with our previous experiments (Fig. 1e, Supplementary Fig. 1e), the addition of DAPT was sufficient to increase the percentage of SAHF-positive IMR90 cells above basal levels both in mono-culture (Fig. 6c) and co-culture (Fig. 6b).

To determine whether tumour cell lines can induce NOTCH1 signalling and repress HMGAs non-autonomously, we repeated the co-cultures and isolated the IMR90 ER:HRAS^G12V mRFP1 cells using flow cytometry (Fig. 6d,

Supplementary Fig. 7b). We found a dramatic upregulation of the canonical NOTCH1 target gene *HEYL* and a concurrent downregulation of *HMGA1* and *HMGA2* in fibroblasts co-cultured with JAG1-expressing tumour cells, particularly A549 and Hep3B cells (Fig. 6e). Two other canonical target genes, *HES1* and *HEY1*, were not dramatically upregulated by JAG1-expressing cell lines (Supplementary Fig. 7c). Although *HEYL*, *HEY1* and *HES1* are known as 'canonical targets' of NOTCH, their transcriptional regulation by NOTCH signalling is highly complex: for example, unique combinations of, or interactions between, NOTCH ligands and receptors can provide preferential induction of certain targets[37,38]. Different tumour cell lines might differentially express other NOTCH ligands (in addition to JAG1) or other NOTCH pathway modulators, conferring additional complexity.

To further determine whether the effect described above is JAG1-dependent, we used CRISPR-Cas9 technology to generate A549 cells with bi-allelic knockout of endogenous JAG1. Two knockout clones were isolated; clone 1 had a 5 bp deletion in the first allele and a 1 bp deletion in the second allele while clone 2 had a 1 bp deletion in the first allele and a 14 bp deletion in the second allele (Supplementary Fig. 7d). A control clone was generated by transfecting cells with Cas9 but omitting guide RNA (−gRNA control cells). We found that both clones 1 and 2 had reduced JAG1 levels by immunoblotting (Fig. 6f) and cell-surface JAG1 (plus JAG2) levels by flow cytometry (Supplementary Fig. 7e). Proliferation was not substantially altered in JAG1-knockout cells relative to control or parental cells (Supplementary

Fig. 7f). In contrast to control or parental A549 cells, co-culture of JAG1-knockout A549 cells with red IMR90 ER:HRAS$^{G12V}$ cells in the presence of 4OHT had little effect on SAHF formation in red (RIS) cells (Fig. 6g).

In addition to JAG1-knockout A549 cells, we generated MCF7 cells containing DOX-inducible JAG1 fused to mVenus (JAG1-mVenus). By immunoblotting, we observed low-level expression of JAG1-mVenus even in the absence of DOX, likely caused by 'leaky' transcription (Supplementary Fig. 7g). Addition of 10 ng/mL of DOX to the culture was sufficient to induce JAG1 to comparable levels as those observed endogenously in Hep3B cells (Supplementary Fig. 7g). Co-culture of MCF7 cells containing inducible JAG1 with red IMR90 ER:HRAS$^{G12V}$ cells was sufficient to reduce the number of SAHF-positive red cells even in the absence of DOX (reflecting the slightly increased levels of JAG1) and completely repress SAHF formation in red cells in the presence of DOX (Supplementary Fig. 7h). While we cannot exclude the effects of other cell-contact-mediated signalling pathways on chromatin structure, our data together demonstrate that JAG1-expressing tumour cells can repress SAHF formation in adjacent senescent cells in a JAG1-dependent manner.

**Non-cell autonomous regulation of chromatin accessibility.** Next, we asked whether tumour cell lines could repress the formation of RIS-driven accessible regions in fibroblasts, as was the case for ectopic N1ICD (Fig. 5b). Utilising flow cytometry, we isolated 4OHT-induced IMR90 ER:HRAS$^{G12V}$ mRFP1 cells after co-culture with tumour cell lines and performed ATAC-seq (Fig. 6d). We found that 66% (29,507), 72% (32,364) and 75% (33,581) of RIS-driven accessible regions were significantly repressed by co-culture with MCF7, A549 and Hep3B cells, respectively (Fig. 6h, i; Supplementary Fig. 8a, b). Co-culture with MCF7, A549 or Hep3B cells induced opening of 6948, 10,303 and 14,064 NIS-driven accessible regions, respectively (Fig. 6h). These data correlated well with the ability of the tumour cell lines to repress SAHFs in adjacent IMR90 (Fig. 6b) and the JAG1 levels expressed by each line (Fig. 6a). RIS-driven accessible regions repressed by co-culture with tumour cell lines overlapped well with each other and with regions repressed by ectopic N1ICD (Supplementary Fig. 8c, d). These data suggest that tumour cells expressing JAG1 can dramatically alter the chromatin landscape of adjacent stromal cells at the nucleosome level.

**HEYL and HMGA1 anti-correlate in multiple tumour types.** If NOTCH1 signalling inhibits HMGA1 in vivo, we would expect an anti-correlation between NOTCH1 activity and HMGA1 expression in human tumour samples. To test this, we first performed a pan-tissue-type analysis using expression microarray data from the R2 database (http://r2.amc.nl) by comparing the expression of HMGA1 and canonical NOTCH1 target genes. When Z-score expression values were analysed in 36,846 human samples, we observed a significant negative correlation between HMGA1 and HEYL ($R = -0.356$, $p < 0.0001$) and HMGA1 and HEY1 ($R = -0.281$, $p < 0.0001$), but no correlation between HMGA1 and HES1 (Supplementary Fig. 9a, b, c). Interestingly, HEYL and HEY1, but not HES1, were also significantly upregulated in IMR90 fibroblasts co-cultured with JAG1-expressing RPE1 cells (Fig. 3g). To study the prognostic importance of this relationship, we used the web-based tool KM-plotter[39,40] and found that patients with low HMGA1 or high HEYL have a significantly better prognosis in lung adenocarcinoma, but not in lung squamous cell carcinoma (SCC) (Supplementary Fig. 9d, e), suggesting that the relationship between these proteins may have prognostic value in certain types of cancer. High HEY1 levels

were prognostic of better overall survival in both types of lung cancer patient (Supplementary Fig. 9d, e).

As microarray data can be dependent on the quality of the probe used, we analysed the co-expression of HMGA1 and HEYL or HEY1 using RNA-seq data generated by The Cancer Genome Atlas (TCGA) Research Network[41] (http://cancergenome.nih.gov). There was a significant negative correlation between HMGA1 and HEYL in the majority of tumour types analysed (Fig. 7a) and a particularly strong anti-correlation in lung SCC (Fig. 7b) ($R = -0.4842$; $p = < 0.0001$). When TCGA patients with lung SCC were categorised based on expression into 'HMGA1 high–HEYL low' and 'HMGA1 low–HEYL high' tumours, patients in the former category had a better overall survival (Fig. 7c) ($p = 0.00316$). We also found a significant negative correlation between HMGA1 and HEY1 in various cancer types, which were not completely overlapping with those where HMGA1 and HEYL anti-correlate (Supplementary Fig. 10a). For example, they were not negatively correlated in lung SCC and the expression of these two genes was not prognostic of patient survival (Supplementary Fig. 10b, c). However, kidney renal clear cell carcinoma showed the strongest negative correlation between HMGA1 and HEY1 and their expression patterns were indicative of prognosis (Supplementary Fig. 10a, d, e). Together, these data demonstrate that an anti-correlation between HMGA1 expression and NOTCH1 activity is evident in cancer and that this correlation can be prognostic of patient outcome.

## Discussion

In the current study, we provide evidence for NOTCH-mediated 'lateral modulation' of chromatin structure at the microscopic and nucleosome scales. While RIS cells form prominent SAHFs at the microscopic scale[13,42], at the nucleosome scale we observed a robust increase in chromatin accessibility. Both SAHFs and RIS-

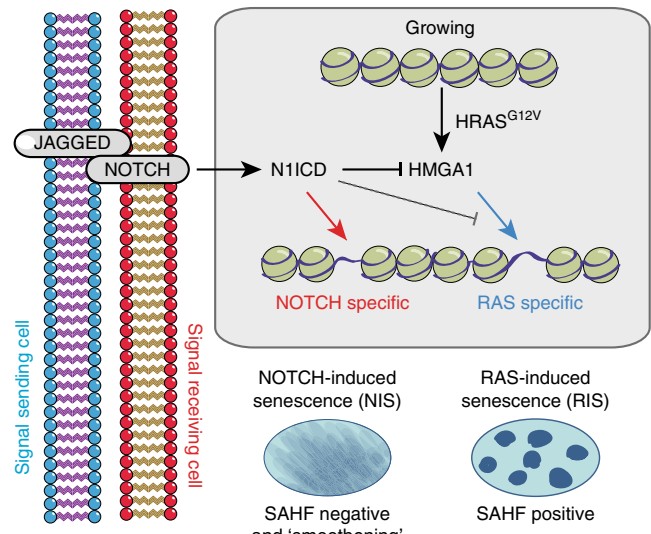

**Fig. 8** NOTCH1 signalling mediates non-cell autonomous regulation of chromatin structure at the microscopic and nucleosome scale. Lateral induction of NOTCH1 activity in a signal-receiving cell by JAG1 on the surface of an adjacent cell (including cancer cells) can drive NIS. NIS cells form unique chromatin-accessible regions and microscopically 'smoothened' chromatin. In the context of RIS, non-cell autonomous activation of NOTCH1 signalling can repress the formation of AT-rich RAS-driven accessible regions at the nucleosome level and SAHF formation at the microscopic level. Mechanistically, N1ICD represses HMGA1, which is responsible for SAHF formation and at least partially for the formation of ectopic-accessible chromatin in RIS cells

driven accessibility can be inhibited by N1ICD-mediated repression of HMGA1 (Fig. 8). While the essential and structural role for HMGA1 in SAHF formation is well established[16], its role in chromatin accessibility is unclear. HMGA proteins compete with Histone-H1 for linker DNA and thus affect chromatin compaction, as demonstrated by techniques such as fluorescence recovery after photo bleaching and MNase digestion assays[43,44]. Our data, using sequencing technology, demonstrate that HMGA1 effects the formation of ectopic accessible regions, potentially by facilitating the binding of other transcription factors, such as C/EBPβ, identified here through motif analysis of RIS-driven accessible regions. It is known that chromatin accessibility is an indicator of developmental maturity[45] and that cancer cells acquire ectopic accessible regions[45,46]. For example, during the metastasis of small cell lung cancer, a dramatic increase in chromatin accessibility at distal regulatory elements allows tumour cells to co-opt pre-programmed gene expression programmes, providing a growth advantage[32]. Thus our data raise a possibility that HMGA1 can drive pluripotency and cancer in part by modulating chromatin accessibility. It will be important to understand how HMGA1 facilitates both chromatin 'opening' at the nucleosome scale and the formation of SAHFs and to determine whether the two are related. We wonder whether the subset of HMGA1-dependent regions that are gene distal could have structural rather than regulatory functionality.

Chromatin accessibility was also increased in NIS cells although these were often at distinct loci. Unlike RIS cells, NIS cells do not form SAHFs and are instead characterised by chromatin 'smoothening'. The mechanisms of chromatin smoothening and formation of NIS-driven accessible regions, and whether these events are related, remains unclear. Note, although knockdown of HMGA1 blocked formation of SAHFs and many RIS-driven accessible regions, it was not sufficient to induce NIS-like chromatin smoothening or NIS-like chromatin accessibility, thus NOTCH signalling modulates chromatin by both HMGA1-dependent and -independent mechanisms (Fig. 5b, c). One possible mechanism through which NOTCH modulates chromatin is through directed histone acetylation[47–50]. N1ICD activates gene transcription by recruiting histone acetyl-transferases[11] and was more recently shown to drive rapid and widespread deposition of H3K56ac[51], which is known to be associated with nucleosome assembly, particularly in DNA replication and repair[52].

NOTCH signalling can be transiently activated during stress-induced senescence (e.g. oncogene- and DNA damage-induced senescence)[9] but also plays important roles during development and in cancer, thus 'lateral induction' of NOTCH activity through JAG1 could affect chromatin structure in various biologically relevant scenarios involving epithelial and/or fibroblast cells. Here we extend our analysis to the more specific 'epithelial-fibroblast' scenario that might mirror the cancer microenvironment where epithelial tumour cells are in active communication with stromal cells through the NOTCH1–JAG1–HMGA1 signalling axis. Consistently, using the *Pten*-null mouse model of prostate cancer, Su and colleagues[53] demonstrated that JAG1 expression in tumour cells facilitates the formation of a 'reactive stroma', which plays an important role in tumour development. It will be important to test whether chromatin structure is altered in the stroma of such tumours and whether this is dependent on HMGA1 repression. In NOTCH-ligand-expressing tumours, targeting chromatin-modifying enzymes in the stromal compartment may present a unique therapeutic opportunity to alter the tumour niche.

## Methods

**Cell culture**. IMR90 HDFs (ATCC) were cultured in Dulbecco's modified Eagle's medium (DMEM)/10% foetal calf serum (FCS) in a 5% $O_2$/5% $CO_2$ atmosphere.

hTERT-RPE1 cells (ATCC) were grown in DMEM-F12/10% FCS in a 5% $O_2$/5% $CO_2$ atmosphere. MCF7, H1299, A549 and Hep3B cells (ATCC) were grown in DMEM/10% FCS in a 5% $CO_2$ atmosphere. Cell identity was confirmed by STR (short tandem repeats) genotyping. Cells were regularly tested for mycoplasma contamination and always found to be negative.

Co-cultures were set-up at a cell number ratio of 1:1 and performed in DMEM/10% FCS in a 5% $O_2$/5% $CO_2$ atmosphere. For transwell experiments, IMR90 cells were plated in the bottom chamber and hTERT-RPE1 or tumour cells were plated into the top chamber of a Corning 12-well Transwell plate (CLS3460 Sigma).

The following compounds were used in cultures: 100 nM 4-hydroxytamoxifen (4OHT) (Sigma), 10 μM DAPT (Sigma), 100 μM etoposide (Sigma), between 10 and 1000 ng/mL doxycycline (DOX) (Sigma) as indicated in individual figures.

**Vectors**. The following retroviral vectors were used: pLNCX (clontech) ER:HRAS^G12V[30]; pWZL–hygro for N1ICD–FLAG (residues 1758–2556 of human NOTCH1[9]) and mRFP1; pLPC-puro for dnMAML1-mVenus (residues 12–74 of human MAML1)[9], mRFP1, rThy1-mRFP1, JAGGED1-mVenus and mVenus; pQCXIH-i for DOX-inducible N1ICD-FLAG[9]; MSCV-puro for miR30 shHMGA1 (shHMGA1 target sequence 5′-ATGAGACGAAATGCTGATGTAT-3′[16]); and pCLIIPi[54] for pCLIIPi JAGGED1-mVenus.

To generate pLPC-puro rThy1-mRFP1, we first PCR cloned mRFP into pLPC-puro (pLPC-puro-x-mRFP, where x denotes cloning sites to express mRFP-fusion proteins). The CDS of rat-Thy1 was PCR amplified from cDNA (a gift from M. de la Roche, CRUK CI, UK), removing the stop codon, before cloning into pLPC-puro-x-mRFP. To generate pLPC-JAGGED1-mVenus, the CDS of human JAGGED1 was amplified using cDNA derived from N1ICD-expressing IMR90 cells, removing the stop codon, before cloning into pLPC-puro-x-mVenus. To generate pCLIIPi (DOX-inducible) JAG1-mVenus, JAGGED1-mVenus was sub-cloned using PCR into pCLIIPi.

**Flow cytometry**. Analysis of ectopic JAG1-mVenus expression was conducted by flow cytometry[9]. Cells were fixed using 4% paraformaldehyde (PFA) in phosphate-buffered saline (PBS) and stained with anti-JAG1-APC (FAB1726A, R&D Systems, 1:10) or isotype control antibody (IC0041A, R&D Systems, 1:10) before analysis on a FACSCalibur flow cytometer (Becton Dickenson). Flow data were further analysed using FlowJo v10.

**MACS and FACS**. MACS of rThy1-expressing cells was performed using CD90.1 microbeads (130-094-523, Miltenyi Biotec) according to the manufacturer's instructions. Fluorescence-activated cell sorting (FACS) was performed using an Influx (Becton Dickenson) flow cytometer.

**Fluorescence microscopy**. Analysis was performed as previously described[16]. Briefly, cells were plated onto #1.5 glass coverslips the day before fixation to achieve approximately 60% confluence. Cells were fixed in 4% (v/v) PFA and permeabilised with 0.2% (v/v) Triton X-100 in PBS with DAPI. Coverslips were mounted onto Superfrost Plus slides (4951, Thermo Fisher) with Vectashield Antifade mounting medium (H-1000, Vector Laboratories Ltd.). Images were obtained using a Leica TCS SP8 microscope with a HC PL APO CS2 1.4NA 100× oil objective (Leica Microsystems). At least 30 nuclei were captured per biological replicate and condition before Fiji[55] was used to calculate nuclear area, standard deviation and maximum intensity of DAPI signal per nucleus. Specifically, the DAPI channel was duplicated, desaturated and a threshold applied using the Otsu method before holes were filled and the 'analyse particles' function was used to create a region of interest per nucleus for measurement in the original DAPI-stained image. SADS were visualised by DNA-fluorescence in situ hybridisation as previously described[15] using fluorescent probes that target the α-satellite repeat sequence (5′-CTTTTGATAGAGCAGTTTTGAAACACTCTTTTTGTA-GAATCTGCAAGTGGATATTTGG-3′). The percentage of SAHF- and SADS-positive cells was counted by scoring at least 200 cells per replicate and condition.

**Quantitative reverse transcription-PCR**. RNA was prepared using the Qiagen RNeasy Plus Kit (74136, Qiagen) according to the manufacturer's instructions and reverse-transcribed to cDNA using the Applied Biosystems High-Capacity Reverse Transcription Kit (43-688-13, Thermo Fisher). Relative expression was calculated as previously described[16] on an Applied Biosystems Quantstudio 6 by the $2^{-\Delta\Delta Ct}$ method[56] using β-actin (*ACTB*) as an internal control. The following primers were used:

*ACTB* forward: 5′-GGACTTCGAGCAAGAGATGG-3′
*ACTB* reverse: 5′-AGGAAGGAAGGCTGGAAGAG-3′
*HEYL* forward: 5′-CTCCAAAGAATCTGTGATGCCAC-3′
*HEYL* reverse: 5′-CCAGGGACAATGAAAGCAAGTTC-3′
*HEY1* forward: 5′-CCGCTGATAGGTTAGGTCTCATTTG-3′
*HEY1* reverse: 5′-TCTTTGTGTTGCTGGGGCTG-3′
*HES1* forward: 5′-ACGTGCGAGGGCGTTAATAC-3′
*HES1* reverse: 5′-ATTGATCTGGGTCATGCAGTTG-3′
*HMGA1* forward: 5′-GAAAAGGACGGCACTGAGAA-3′
*HMGA1* reverse: 5′-TGGTTTCCTTCCTGGAGTTG-3′
*HMGA2* forward: 5′-AGCGCCTCAGAAGAGAGGA-3′

*HMGA2* reverse: 5′-AACTTGTTGTGGCCATTTCC-3′

**Protein quantification by immunoblotting.** Immunoblotting was performed using sodium dodecyl sulphate-polyacrylamide gel electrophoresis gels using the following antibodies: anti-β-actin (Sigma, A5441, 1:10,000); anti-HRAS (Calbiochem, OP-23, 1:500); anti-NOTCH1 (Cell Signaling, 4380, 1:500); anti-HES1 (Cell Signalling, 11988, 1:1000); anti-FLAG (Cell Signaling, 2368, 1:1000), anti-HMGA1 (Cold Spring Harbor Labs, #37, 1:1000); anti-HMGA1 (Abcam, Ab4078, 1:1000); anti-HMGA2 (Cold Spring Harbor Labs, #24, 1:1000); anti-GFP (Clontech 632377, 1:1000); and anti-JAG1 (Cell Signaling, 2155, 1:1000). Images of uncropped immunoblots are included in Supplementary Fig. 11.

**ATAC-seq.** ATAC-seq samples were generated as previously[31] using 100,000 IMR90 cells and 13 cycles of PCR amplification. Samples were size selected between 170 and 400 bp (in order to isolate 'nucleosome free' and 'mono-nucleosome' fragments) using SPRIselect beads (B23319, Beckman Coulter) before single-end sequencing to generate 75 bp reads on the NextSeq-500 platform (Illumina).

**ChIP-seq.** Chromatin immunoprecipitation (ChIP) was performed as previously described using 20 μg of sonicated chromatin[57] from growing and RIS IMR90 ER:HRAS$^{G12V}$ cells and 5 μg of anti-H3K27ac antibody (Clone CMA309[58]) or 5 μg of H3K4me1 antibody (Clone CMA302[58]). Libraries were prepared using the NEB-Next Ultra II DNA Library Prep Kit for Illumina (37645, New England Biolabs) according to the manufacturer's instructions except that size selection was performed after PCR amplification using SPRIselect beads (B23319, Beckman Coulter). Samples were sequenced single-end using 50 bp reads on the HiSeq-2500 platform (Illumina).

**RNA-seq.** RNA-seq data was generated from IMR90 ER:HRAS$^{G12V}$ cells expressing a short-hairpin targeting the 3′ UTR of human *HMGA1* (RIS+shHMGA1). RNA was purified as above and quality checked using the Bioanalyser eukaryotic total RNA nano series II chip (Agilent). mRNA-seq libraries were prepared from six biological replicates of each condition using the TruSeq Stranded mRNA Library Prep Kit (Illumina) according to the manufacturer's instructions and sequenced using the HiSeq-2500 platform (Illumina).

**Generation of genome-edited JAG1 knockout clones.** The following CRISPR guides were designed against Exon 2 of *JAG1* (NM_000214.2) (Supplementary Fig. 7d):

sgJAG1_2.1: 5′-AGTCCCGCGTCACGGCCGGG-3′ (PAM:GGG) and
sgJAG1_2.2: 5′-CGCGGGACGTGATACTCCTTG-3′ (PAM:AGG).

Oligonucleotides (Sigma Aldrich) were cloned into pSpCas9(BB)-2A-GFP[59]. pSpCas9(BB)-2A-GFP (PX458) was a gift from Feng Zhang (Addgene plasmid # 48138). Guide cutting efficiency was determined in A549 cells using the T7 assay (New England Biolabs, following manufacturer's instructions). To generate independent, non-sister clonal cell lines, A549 cells were transiently transfected (Lipofectamine 3000, Thermo Fisher Scientific) with PX458-empty (control), PX458-sgJAG1_2.1 and PX458-sgJAG2.2, and single cell was cloned 96 h post-transfection by FACS (BD FACSAria II). gDNA was extracted from each clone (Extracta DNA Prep, VWR, 95091-025) and Exon 2 of JAG1 was amplified by PCR (FastStart HF System (Sigma Aldrich, 3553361001)) using the following primers (universal Fluidigm tag in lower case, JAG1-specific sequence in upper case)):

Forward: 5′-acactgacgacatggttctaca-GAGCTGCAGAACGGGAACT-3′;
Reverse: 5′-tacggtagcagagacttggtct-CTTGAGGTTGAAGGTGTTGC-3′.

Amplicons were diluted 1:150 and re-amplified with Fluidigm barcoding primers (incorporating a unique sample barcode and Illumina P5 and P7 adapter sequences), pooled and subjected to sequencing (Illumina MiSeq platform). The AmpliconSeq analysis pipeline was used for data processing and variant calling. Briefly, reads were aligned against the reference genome (GRCh38) using BWA-MEM[60] and variants were called using two methods (VarDict[61] and GATK HaplotypeCaller[62]). Consensus variants and their effects on CRISPR clones were then calculated. All clones used in this paper were STR genotyped and confirmed as free from mycoplasma.

**RNA-seq analysis.** Reads were mapped to the human reference genome hg19 with the STAR (version 2.5.0b) aligner[63]. Low-quality reads (mapping quality <20) as well as known adapter contaminations were filtered out using Cutadapt (version 1.10.0)[64]. Read counting was performed using Bioconductor packages Rsubread[34] and differential expression analysis with edgeR[33,34]. The conditions were contrasted against the growing samples. Genes were identified as differentially expressed with a FDR (false discovery rate) cut-off of 0.01 and an absolute value of logFC (log$_2$ of the fold change) >0.58.

**ChIP-seq and ATAC-seq analysis.** ChIP-seq and ATAC-seq reads were mapped to the human reference genome (hg19) with BWA (version 0.7.12)[60]. Low-quality reads (mapping quality <20) as well as known adapter contaminations were filtered using Cutadapt (version 1.10.0)[64], and reads mapping to the 'blacklisted' regions identified by ENCODE[65] were further removed. Average fragment size was

determined using the ChIPQC Bioconductor package[66], and peak calling was performed with MACS2 (version 2.1.0)[67], using fragment size as an extension size (--extsize) parameter. High-confidence peak sets for each condition were identified separately using only those peak regions that were present in at least two replicates.

**Differential accessibility analysis.** THOR[35] and edgeR[33,34] were used to identify differentially accessible regions between conditions. For the comparisons 'NIS vs. growing' and 'RIS vs. growing', the intersect of regions detected by THOR and edgeR was taken. This approach gave us a robust set of regions that are altered in RIS and NIS conditions. For other comparisons where volcano plots were generated, edgeR was used to interrogate how different genetic and cell-culture manipulations effect the alterations detected in RIS.

edgeR[33,34] was used on a merged set of growing, RIS, NIS, N+RIS, shHMGA1 and RIS/shHMGA1 high-confidence ATAC-seq peak sets (present in at least two replicates of a single condition) to identify regions of differential accessibility between conditions. We utilised the TMM method implemented in edgeR for normalisation and dispersion calculation of the replicated samples. The results were further filtered based on FDR < 0.05 and logFC ≥ 0.58 or ≤−0.58.

THOR is a Hidden Markov Model based approach that utilises all mapped reads and identifies the differentially accessible regions between two conditions. THOR was used in parallel with edgeR to identify the differences between the conditions using our pre-computed normalisation factors (see section 'Generation of normalised coverage files') to normalise between samples. Regions were further filtered using a −log(p-value) cut-off of 10.

**Annotating differentially accessible regions.** Bedtools intersect (v2.26.0)[68] was used to identify regions annotated as 'unchanged' by extracting high-confidence peaks in the 'growing' condition that did not intersect with regions that become more or less accessible in the NIS and/or RIS conditions (relative to the growing condition). The rest of the categories (open or closed in RIS or NIS) were identified based on the differential accessibility analysis compared to growing using the intersect of THOR and edgeR as described above. Regions displaying altered chromatin accessibility were mapped to genomic annotations or repeat regions using bedtools v2.26.0[68]. For the genomic annotations, we used TSSs from the FANTOM database[69], repeats from repeatMasker (UCSC genome browser) and other genomic features (exons, introns, UTRs, etc.) were extracted from the UCSC Table Browser. The enhancers were identified based on our own H3K4me1 and H3K27ac histone mark ChIP-seq data sets; all regions that had peaks in both of these marks in either growing or RIS cells were considered as enhancers.

**Intersecting consensus peaks and generating Venn diagrams.** The Homer (v3.12)[70] command 'mergePeaks' with default settings and the output options '–venn' and '–prefix' were used to generate values for plotting Venn diagrams and associated bed files for further analysis. Only literal overlaps (overlapping by 1 bp) were considered. Venn diagrams were plotted using the R package 'Venneuler' (https://cran.rproject.org/web/packages/venneuler/index.html).

**Calculating proximity to genes and GC percentage.** To calculate the distance of consensus peaks from TSSs and GC percentage of accessible regions, the Homer (v3.12)[70] command 'annotatePeaks.pl' was used with default settings and the output option '-CpG'.

**Gene enrichment analysis.** Altered accessible regions within 500 bp of a gene TSS were identified using Homer as described above. Gene enrichment analysis was performed using the GO Biological Process 2015 annotation provided on the web-tool 'Enrichr'[71] (http://amp.pharm.mssm.edu/Enrichr/).

**Generation of normalised coverage files.** A previously described approach was used to generate scaling factors for each ATAC-seq condition relative to others[32]. Briefly, we reasoned that the enrichment of reads within ATAC-seq peaks containing TSSs of genes that are both expressed (logCPM > mean logCPM) and have low variance between conditions (−0.14 < logFC < 0.14) by RNA-seq should not vary, unless there are differences in ATAC-seq sample quality, preparation or sequencing. By reanalysing our previously published IMR90 RNA-seq data[9] together with newly generated RNA-seq samples for RIS+shHMGA1 cells, we identified 589 genes that fit these criteria. We counted the reads from the ATAC-seq samples that map to these specific genes using Rsubread[34] and computed scaling factors based on the mean counts for each condition separately. Normalised coverage files (bigWig) were generated by pooling reads from all of the replicates and applying the calculated scaling factors using the 'genomecov' function in bedtools, sorting the resulting normalised bedGraph files and then converting them to bigWigs using the 'bedGraphToBigWig' function from UCSC.

**Generation of clustered heatmaps.** Heatmaps were generated using normalised coverage of peaks (+/−2.5 kb) representing novel accessible regions (regions with significantly altered chromatin accessibility in RIS or NIS relative to growing cells) with k-means clustering using the deepTools package[72].

**PCA analysis and correlation heatmaps**. Samples were normalised with the pre-calculated normalisation factors (as described above in 'Generation of normalised coverage files'), and reads from all growing, RIS, NIS, N+RIS, shHMGA1 and RIS/shHMGA1 consensus peak sets (present in at least two replicates across all of the samples) were extracted and used in the PCA analysis and the correlation analysis of data sets. Pearson correlation was calculated between samples based on these normalised read counts and correlation heatmaps were generated with pheatmap (http://CRAN.R-project.org/package=pheatmap) and WPGMA clustering. PCA plots were generated using ggplot2[73].

Reads from publicly available ATAC-seq and DNase-seq data sets[74–77] (references and NIH Epigenomics Roadmap Initiative) were extracted from the same regions; however, since these were not included in normalisation factor calculation, standard CPM normalisation was used for Supplementary Fig. 4c.

**Volcano plots**. edgeR calculated statistical parameters (logFC and logFDR) were used to visualise differentially accessible regions in RIS (red) and NIS (purple) compared to growing cells in the comparisons indicated. Plots were generated using ggplot2[73].

**Motif enrichment analysis**. Meme-ChIP suite (version 4.12.0), together with Hocomoco (version 11) human and mouse PWMs, was used to detect motif enrichment in a 600 bp region centred at the peak summit.

**TCGA analysis**. We analysed the expression levels of NOTCH-associated genes in the publicly available RNA sequencing data generated by the TCGA Research Network: http://cancergenome.nih.gov/[41].

Computational analysis and statistical testing of the Next-Generation Sequencing data was conducted using the R statistical programming language[78]. Filtered and log$_2$-normalised RNA expression data along with all available clinical data were downloaded from the GDAC firehose database (run: stddata_2015_06_01) for each gene of interest from the relevant cancer-specific collections.

Correlation testing for associations between expressed genes was performed using the cor.test function in R to calculate the Pearson's product moment correlation coefficient and test for significant deviation from no correlation. Plotting of TCGA data was performed using the ggplot2 R package[73]. Survival analysis was performed using the survminer and survival[79] R packages. Kaplan–Meier estimated survival curves were constructed using the TCGA clinical data. Statistical testing of differences between survival curves used the G-rho family of tests, as implemented in the survdiff function of the survival package.

**Statistics and reproducibility**. No statistical method was used to predetermine sample size and experiments were not randomised. Statistical analyses were conducted using the Graphpad Prism 6 and R statistical software, except for TCGA data analysis (which was as described in the methods above). One-way analysis of variance with Tukey's correction for multiple comparisons was used for data sets with >2 conditions. Two-sample $t$-tests were used for two-condition comparisons. The statistical tests were justified as appropriate based on the number of samples compared and the assumed variance within populations. A $p$-value of <0.05 was used to indicate statistical significance.

**Data availability**. The RNA-seq, ChIP-seq and ATAC-seq data generated for this study have been deposited at the Gene Expression Omnibus (GEO) with the accession number GSE103590. Gene expression data from RIS, NIS and N+RIS is previously published[9] and available under GEO accession number GSE72404.

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

## Acknowledgements

We thank all members of the Narita laboratory for helpful discussions, M. de la Roche for reagents and staff of the Cancer Research UK Cambridge Institute core facilities for technical support. The University of Cambridge, Cancer Research UK and Hutchison Whampoa supported this work. M.N., S.B. and I.A.R laboratories are funded by a Cancer Research UK Cambridge Institute Core Grant (C14303/A17197). M.N. is also supported by a Cancer Research UK Early Detection Pump Priming award (C20/A20976), Medical Research Council (MR/M013049/1) and Tokyo Tech World Research Hub Initiative (WRHI). M.H. is supported by a CRUK Clinician Scientist Fellowship (C52489/A19924). R.H.-H. is funded by an EMBO Long-Term fellowship. S.A.S. and D.B. were supported by Medical Research Council core funding. H.K. was supported by JSPS KAKENHI JP25116005, JP26291071, 15K21730 and 17H01417.

## Author contributions

A.J.P., M.H. and R.H.-H. designed experiments, performed experiments and analysed data. D.B., S.A.S. and A.J.P. analysed sequencing data. S.S. analysed TCGA data. E.M., A.S., P.D'S and I.A.R. generated A549 JAG1 knockout cell lines. H.K. provided the antibodies for H3K27ac and H3K4me1 ChIP-seq. S.B., S.A.S., H.K. and M.N. supervised experiments and interpreted the data. M.N., M.H. and A.J.P. conceived the project. M.N. and A.J.P. wrote the manuscript with contribution and review from all authors.

## Additional information

**Competing interests:** The authors declare no competing interests.

