## [Peer Review File · Nature Communications]

Reviewers' comments:

Reviewer #1 (Remarks to the Author):

Review NCOMMS-17-26568-T

"NOTCH-mediated non-cell autonomous regulation of chromatin structure during senescence"

In this manuscript the Narita group analyzes changes in chromatin structure during senescence and the role of NOTCH signaling in the process. The group demonstrates that senescence induced by expression of the NOTCH intracellular domain is dominant over Ras induced senescence and that NOTCH signaling in that context inhibits chromatin condensation into SAHF by a so called smoothing effect. This effect could be communicated to other cells via Jag1 signaling. Notch suppression strongly suppressed HMGA expression, suggesting that this was the mechanism of how NOTCH affected SAHF formation.

ATAC seq revealed important differences between replicative senescence and RIS or NIS, which displayed global chromatin flexibility and opening at the nucleosomal level, which in turn was reflected in the transcription signature. The dominance of NIS over RIS was also reflected in these approaches, as nucleosome free regions in RIS were abrogated by NOTCH.

Extending their work to cancer cells, the group demonstrates convincingly that also in a tumor cell system JAG1 alters the chromatin landscape of adjacent cells, which is supported by the finding of an inverse correlation between HMGA1 and NOTCH activity.

In summary this is an elegant study that brings many novel findings to the table and puts it into the context of senescence and chromatin organization in senescent cells as well as in cancer. The discoveries about the JAG1-HMGA1-SAHF pathway required for cell communication and chromatin organization in adjacent cells is fascinating.

The experiments are planned and performed extremely well. It would be important to clarify the PI's thoughts about differences in fibroblasts and epithelial cells, considering both are used for the cell communication experiments in Figure 3, the fact that epithelial cells give rise to cancers much more frequently than fibroblasts, and that fibroblast-senescence is much more stringent than epithelial cell senescence.

Furthermore, do the transcriptional signatures observed with NIS reflect changes observed in aging cells?

Considering the technical quality of the work, the interest for a wide audience and the important scientific advances presented in the manuscript, it is very well suited for publication in Nature Communications.

Reviewer #2 (Remarks to the Author):

Parry et al. present a manuscript which is a natural extension of their recent publication in Nature Cell Biology (Hoare et al., 2016) in which 'lateral induction' of senescence is proposed as a novel route to senescence. The authors employ the following models of senescence: RAS-induced senescence (RIS; IMR90 ER:RAS) and Notch-induced senescence (NIS). RIS is associated with the induction of senescence-associated heterochromatic foci (SAHFs) and an increase in nucleosome-free regions (NFRs; ATAC-Seq). By contrast, NIS occurs in the absence of SAHFs, and can act to suppress SAHFs and NFRs in a RIS context. The authors propose that non-autonomous NOTCH 1 signaling and JAG1 (JAGGED1, Notch ligand) expression repress SAHFs in adjacent cells, in part by repression of the chromatin architecture proteins HMGA1 and 2. The authors perform ATAC-sequencing, ChIP-seq for the enhancer-associated modification H3K27Ac and GO analysis for RIS, NIS, N+RIS and growing IMR90 cells to identify NFRs. This is combined with ectopic N1ICD and HMGA1 knockdown to assess the influence on RIS-specific NFRs. The authors then propose that tumour cells 'repress' SAHFs and RIS-specific NFRs in a cell-contact dependent manner through the use of three cancer cell lines selected for low, medium and high JAG1 expression. Finally, data

from TCGA and KMPlot is presented which indicates an anti-correlation between HEY-L (canonical NOTCH1 target gene) and HMGA1 in a panel of tumour types, and that HEY-L-Low/HMGA1-High expression shows increased survival in lung-squamous cell carcinoma. Taken together, this work highlights a role for HMGA1 in RIS-specific NFR formation.

The manuscript was a pleasure to read, is well written with a good logic flow and generally sound statements and conclusions. The discovery of differences between RIS and NIS NFRs is novel, and considerable effort has been taken to explore the JAG1-HMGA1 axis at a global level. Detailed below are some issues and requests for additional details which would help improve the manuscript further.

Major Issues

Figure 6 and Supplementary Figure 6.

MCF7, A549 and Hep3B cells are selected based on their JAG1 protein expression (low, medium and high, respectively, Figure 6a). Co-culture of IMR90 ER:RAS results in a decrease in the percentage of SAHF positive nuclei which is inversely related to the JAG1 expression (Figure 6b), and similar results are presented for HMGA1 and HMGA2 mRNA levels (Figure 6d). However, there seems to be a disconnect between JAG1 levels and the impact in co-culture experiments on HEY-L and HES1 levels (Supplementary Figure 6c), i.e. they do not directly relate to JAG1 levels, which suggests that the effects that are being attributed to JAG1 may be an oversimplification. This internal inconsistency is skirted over within the text and should be addressed more specifically.

The authors perform transwell experiments with MCF7/A549/Hep3B cells and IMR90 ER:RAS which indicate that co-culture is important for repression of SAHF formation (Supplemental Figure 6b) and co-culture in the presence of DAPT (γ -secretase inhibitor which inhibits downstream signaling of NIS) (Figure 6b). However, given that these three cancer cell lines are likely to be different many other potential cell surface proteins in addition to JAG1, the authors should attempt a co-culture experiment combined with specific JAG1 knockdown and determine the impact on SAHF positive cells to establish that this is a JAG1-mediated effect. It may be that the authors have attempted such a set of experiments and observed reduced viability with JAG1. If so, it would be helpful to see such data. Alternatively, inducible (and therefore titratable) expression of JAG1 could be an alternative route to assess this in an isogenic manner.

Minor Issues

1. The work presented within Figure 1 raises the question as to whether ectopic expression of N1ICD in established RIS can alter SAHF formation once they have been established, and if so, shift the senescence state from RIS to NIS. In other words, is there plasticity within the RIS phenotype (SAHFs and NFRs) or is this phenotype 'locked in' once SAHF are established.

2. Page 11 Line 20 onwards. Can the authors provide the details for how many peaks scored for 3/3 samples for each of the NFRs.

3. Page 12 Line 20 onwards. Is the data presented for intersected consensus peaks for 2 out of 3 samples again?

4. Page 13 Line 5 onwards. NFRs within 300bp of a gene TSS were taken forward for GO analyses and the authors describe common pathways with previous RNA-seq data. Can they provide details of the specific genes which were identified by this analysis and of these which are specifically in common with the RNA-seq data. i.e. do the genes which alter in the RNA-seq data also have altered NFRs, or is it a 'coincidental' overlap in pathways which is being reported.

5. Page 13 line 16. Since this manuscript was submitted there have been further publications which explore the role of pro/anti-apoptotic pathways. Perhaps the authors could extend their bibliography to include these papers.

6. Page 14 Line 18 onwards. Was ATAC-seq performed for both IMR90 ER:RAS and IMR90 ER:RAS+shHMGA1 or just the later. If ATAC-seq was performed for both samples have does this data compare to the ATAC-seq data presented earlier in the manuscript?
7. Page 15 Line 20 onwards. Did the authors perform ATAC-seq for NIS+shHMGA1, and is so what did this data support their assertion that NIS occurs via an HMGA-independent mechanism.
8. Page 17 Line 7 onwards. The authors ask whether tumour cell lines could repress RIS-unique NFR formation in IMR90 ER:RAS. What is the overlap with the ATAC-seq performed in this paragraph with NIS NFRs and N+RIS NFRs? How does this compare with the intersections for the cancer cell lines with increasing JAG1 expression.
9. Page 18 Line 10 onwards. I think that microarray data for just 3,554 breast cancer patients in presented within the figure, rather than the full 5,143 as stated. Likewise, I think data for just 1,145 lung cancer patients is shown. Please provide details as to whether is RFS, OS etc.
10. Page 18 Line 4. Should this read ' $r=-0.4842$ ' rather than ' $r=0.4842$ '.
11. Page 22 Line 4 onwards. Please provide the dose(s) of each compound used within this manuscript.
12. Page 24 Line 2. Please provide detailed information as to how a SAHF and SADS positive cell (nuclei?) were defined as positive versus negative. In some of the images it looks as if there may still be a small number of SAHFs present, and it is unclear how this criteria has been applied.
13. Page 25. Please provide details of validation of the anti-H3K27Ac and anti-H3K4me1 antibodies prior to ChIP-seq. Was any post ChIP-seq validation performed?
14. Page 27 Line 17 onwards. "Enhancers were identified based on our own H3K4me1 and H3K27ac ...ChIP-seq datasets" – are these published, if so please provide a reference. Alternatively, how have these been validated to ensure that enhancers were accurately identified.

Figure 1.

Please provide more technical details of how the experiment was performed, i.e. at what time point is OHT added etc.

Figure 1b. Is the image of the RAS nucleus representative?

Figure 1e. As per notes above, please provide details of the DAPT dose employed. The level of SAHFs at time zero for the mVenus-dnMAML1 appears to be significantly (**) elevated relative to the mVenus control. Please provide an explanation for this. Please also provide a representative image of the mVenus + DAPT.

Figure 2e. From the images presented, it looks as if the RPE1 JAG1 mVenus nuclei (Blue) have are SAHF positive. Can RAS drive SAHF formation in adjacent RPE1 nuclei?

Figure 5c. Do the width of NFRs alter in RIS+shHMGA1 relative to RIS, especially in cluster 1. If so, what is the authors interpretation of this data?

Figure 6b. How does this data relate to the data in Supplemental Figure S1E? What does of DAPT was used? Please provide the Alone +DAPT data for direct comparison and relate this to the findings with the co-culture with the cancer cell lines.

Supplemental Figure 4b. The PCA analysis indicates that NIS-1 and N+RIS-1 do not cluster with their corresponding -2 and -3 samples. Were both of these excluded from future analysis? See also

text on page 11 Line 15.

Figure legends

Please add details of compound doses used within a given experiment for clarity.

Consistency.

Please check the manuscript for general consistency, e.g. IMR90 ER:RAS or IMR90 ER:HRAS, HEYL or HEY-L.

Reviewer #3 (Remarks to the Author):

In this paper, Parry and colleagues report some mechanistic observations differentiating NOTCH-induced from RAS-induced senescence, especially on how juxtacrine signaling might induce a chain of events leading to large chromatin re-organization. They also show the level of expression of some of these genes to be predictive of survival of lung and breast cancer patients.

I have very limited knowledge of cellular senescence and this field in general, so I am not able to give a proper feedback about the novelty of the manuscript. Overall, I found it easy to follow (except for a number of acronyms and use of jargon) and providing interesting observations whose relevance for primary tumors is computationally validated using publicly available TCGA datasets. Nevertheless, I think the analyses of the generated genome-wide profiles of chromatin accessibility are affected by some major flaws that currently mine one of the main conclusions of the paper, namely chromatin remodeling at tens of thousands of loci in senescent cells. These shortcomings must be addressed in order to justify publication of the manuscript in Nature Communications.

The authors made a genuine effort in generating multiple replicates of the ATAC-seq libraries per condition, but a larger analytical effort is required to support the authors' conclusions. The results of the functional analyses (e.g. Fig. 4F) are promising and supported by a few anecdotal screenshots provided in Fig. 4 and Supp. Fig. 5. Nevertheless, these need to be confirmed by more thorough analyses of the data (see major comments).

###

Major comments:

1. The major difference in the total number of accessible regions seen between the growing and the RIS/NIS/R+N samples (Fig, 4D) is currently not strongly supported by data analyses. While the results seem to point to a strong qualitative difference in the number of open-chromatin regions, the way the analyses were performed raises a number of questions on how this difference would be rather quantitative than qualitative.

1A. Most of the peaks might have not been identified in the growing samples given that only 1 out of 4 replicates for that condition showed a signal-to-noise ratio (considering the RIP% as a surrogate for that) in line with most of the samples in the other conditions (Supp. Fig. 4A). Given the authors required a peak to be present in at least two samples, separately for each condition, this clearly put the growing samples at a disadvantage in terms of sensitivity.

1B. As a quality control, the authors should run MACS using increasingly more lenient thresholds. If the number of peaks called between the growing and the other conditions converges to more similar numbers, this would be a good indication that the difference is mainly driven by the signal-to-noise issue highlighted above.

1C. In any case, the authors should use a more appropriate approach to quantify the differences in accessibility between conditions. A standard way of doing this is to apply approaches such as edgeR (PMID: 19910308) or DEseq (PMID: 20979621) starting from the normalized counts of the identified peaks. These methods have been conveniently wrapped into a Bioconductor package

called DiffBind.

1D. Given the limitations of the current analyses, the results shown in Supp. Fig. 4C-E cannot be fully evaluated. Besides, given the standard alignment pipeline only considers uniquely mappable reads, the repetitive element identified might actually be only the tip of the iceberg of a much larger phenomenon, which would require the analysis of multi-mapping reads (PMID: 22124482). This should be at least acknowledge if not investigated.

2. A description on how the PCA showed in Supp. Fig. 4B was performed is lacking. The first principal component seems to separate the samples by RIP%. This is in fact showing that there is more variability between NIS replicate 1 and N+R replicate 1 and all the other replicates in these to conditions, rather than compared to the growing samples. This would suggest excluding these samples from any downstream analyses, unless the PCA was performed using the significance level of peaks rather than the normalized signals (if this is the case, the PCA should be repeated).

3. Similarly, a description on how the clustering in Fig. 4C was carried out is lacking and should be provided.

4. The "RIS+shHMGA1" ATAC-seq sample should also be included in the PCA analysis. It should be quality controlled (e.g. what is the RIP for this sample?) and processed as suggested above for the other samples (e.g. by directly comparing the signals from the RIS+shHMGA1 condition with the RIS, Fig. 5A). The same considerations apply to the samples shown in Fig. 6.

5. The conclusion that "HMGA1 is a key regulator of chromatin structure at the nucleosome level" (page 15) is currently not fully supported by data analysis. Given binding-motifs for HMGA1 have been published (for example see the collection HOCOMOCO: <http://hocomoco11.autosome.ru/>), the authors should at least perform an over-representation analysis for known transcription-factor-binding-motifs in the different groups identified in Figure 5. This unbiased analysis might confirm that the HMGA1 motif is indeed highly enriched in the RIS-unique group and importantly, might point to other transcription factors that could differentially bind these regions.

6. While the analysis in Fig. 5C is in principle well-done, there seems to be an issue with visualization (it might simply be a problem with the pdf). Some of the boxes are completely white, and in some of them a small version of the corresponding heatmap seems to appear in the bottom-left corner.

7. One of the major points raised in the first part of the paper is the massive downregulation of HMGA genes in NOTCH-induced senescence (Fig. 3). The authors should look at the genomic landscapes of HMGA1 and HMGA2 in the ATAC-seq profiles they generated. This would at least provide further support for their conclusions (are the promoter and enhancers around these genes showing decreased accessibility?), if not to more a precise mechanistic hypothesis.

####

Minor comments:

1. The authors show that both HEY1 and HEYL are significantly (although slightly) anti-correlated with HMGA1 in terms of expression in cancer (Supp. Fig. 7). Despite that, they pick only HEYL because it is the most up-regulated in the experiments shown in Figs. 3 and Figs. 6. Nevertheless, HEY1 is only slightly less up-regulated than HEYL (Fig. 3F, but in a more robust way than HEYL) or not shown at all (Fig. 6). I think the authors should at least comment on (if not show) the results of running similar analyses to those shown in Fig. 7 and Supp. Fig. 8 for HEY1.

2. Page 11: "When interrogated using a genome browser, these were comparable to each other and to ENCODE DNase sequencing data from normal human lung fibroblasts (NHLF)". This is a good starting point but it has to be shown more formally, e.g. by doing a genome-wide correlation

analysis between the ATAC-seq samples and a few published DNase profiles, including the NHLF and a few unrelated ones.

3. Fig. 4, the authors should provide scale for the y-axis when reporting screenshots from genome browsers (such as UCSC).

4. In the methods section, page 27: "A similar filtering was carried out using Cutadapt as 5 described for RNA-seq data". This step of the analysis is unclear and needs to be clarified.

5. In the abstract: NRFs = NFRs?

6. The acronym SADS appears on page 6 for the first time and is never spelled out.

We thank the reviewers for their thoughtful comments and the editors for their guidance; we feel that the manuscript has been significantly improved as a result. Below we have responded to comments point-by-point:

Reviewer #1:

In this manuscript the Narita group analyzes changes in chromatin structure during senescence and the role of NOTCH signaling in the process. The group demonstrates that senescence induced by expression of the NOTCH intracellular domain is dominant over Ras induced senescence and that NOTCH signaling in that context inhibits chromatin condensation into SAHF by a so called smoothing effect. This effect could be communicated to other cells via Jag1 signaling. Notch suppression strongly suppressed HMGA expression, suggesting that this was the mechanism of how NOTCH affected SAHF formation. ATAC seq revealed important differences between replicative senescence and RIS or NIS, which displayed global chromatin flexibility and opening at the nucleosomal level, which in turn was reflected in the transcription signature. The dominance of NIS over RIS was also reflected in these approaches, as nucleosome free regions in RIS were abrogated by NOTCH. Extending their work to cancer cells, the group demonstrates convincingly that also in a tumor cell system JAG1 alters the chromatin landscape of adjacent cells, which is supported by the finding of an inverse correlation between HMGA1 and NOTCH activity.

In summary this is an elegant study that brings many novel findings to the table and puts it into the context of senescence and chromatin organization in senescent cells as well as in cancer. The discoveries about the JAG1-HMGA1-SAHF pathway required for cell communication and chromatin organization in adjacent cells is fascinating. The experiments are planned and performed extremely well.

1. It would be important to clarify the PIs thoughts about differences in fibroblasts and epithelial cells, considering both are used for the cell communication experiments in Figure 3, the fact that epithelial cells give rise to cancers much more frequently than fibroblasts, and that fibroblast-senescence is much more stringent than epithelial cell senescence.

This is an important point. We have previously shown that NOTCH/JAG1 signalling can be transiently activated during stress-induced senescence (e.g. oncogene-induced and DNA damage-induced senescence) and described the possibility that the transient activation of NOTCH signalling may dynamically fine-tune the senescence program¹. This is not unique to fibroblasts but could apply to other cells including epithelial cells, particularly the type of cells in which NOTCH activates 'lateral induction' (i.e. the NOTCH activation induces expression of JAG, which activates NOTCH signalling in the adjacent cells). The present study suggests that non-

autonomous epigenetic regulation might be involved in this process. To simplify this complex system, we have used NOTCH1/JAG1-expressing cell models, capturing the 'high-NOTCH phase' of senescence. To comprehensively understand the dynamic nature of non-autonomous (juxtacrine) activities of NOTCH/JAG1 signalling would require single cell and systems biology approaches. This would be an excellent future direction.

In addition to this general (either epithelial or fibroblast) aspect of NOTCH signalling in senescence, we applied the idea (of non-cell autonomous epigenetic regulation) to the more specific 'epithelial – fibroblasts scenario'. As the reviewer correctly points out, most cancers are derived from epithelial cells and those cells are actively communicating with stromal cells, including fibroblasts. Indeed, functional interaction between JAG1-expressing cancer cells and stromal cells has been reported ², and our data potentially provide a mechanistic insight into this. In this case, epithelial cells (cancer or perhaps pre-cancerous cells) serve as signal sending cells. In addition to the cancer context, such cell contact-mediated epigenetic regulation might also be involved in embryonic development, where NOTCH/JAG1 signalling (lateral induction of NOTCH signalling) is known to play a critical role (this possibility is not tested in the current study).

We have now extended our discussion to incorporate this point (page 22, line 20 onwards).

2. Furthermore, do the transcriptional signatures observed with NIS reflect changes observed in aging cells?

Whether or not NIS plays a role in aging is an interesting question given recent work implicating senescent cells in age-associated disease. We first performed gene-set enrichment analysis (GSEA) using MSigDB curated gene sets on our own NIS versus growing mRNA-seq data ¹. Within the gene-set collection, 'chemical and genetic perturbations', which includes a number of ageing-associated gene sets, we did not observe substantial enrichment of age-associated upregulated genes in NIS.

We next conducted a similar analysis in a culture model to examine if cells undergoing replicative senescence ('old') develop a NIS-like transcriptome. We reanalyzed publically available mRNA-seq data generated from old and young IMR90 cells ³ (GSE63577). First, we identified the top 100 up-regulated and top 100 down-regulated genes in old cells relative to young cells (based on log fold-change). We used these gene lists and our NIS versus growing mRNA-seq data to perform GSEA. As expected, downregulated genes in old cells were also downregulated in NIS (Reviewers Fig. R1a): these genes were mostly cell cycle genes (Reviewers Fig. R1b). In contrast, the upregulated gene set in old cells was not significantly increased in NIS (Reviewers Fig. R1b). These analyses suggest that a NIS-like transcriptome is not evident in

aging/old cells.

It remains possible that a NIS-like transcriptome is temporally regulated during the initiation of the replicative senescence program, as we have recently demonstrated in more acute models of senescence (RAS- as well as DNA damage-induced senescence) ¹. It is also possible that replicative senescent cells are more heterogeneous, containing both conventional and NIS-type senescent cells, Indeed, the GSEA analysis (Reviewers Fig. R1a) shows upregulated genes in replicative senescent cells appear to be enriched in both up- and down-regulated genes in NIS cells. Note, up-regulated genes in RIS or DNA damage-induced senescence are often downregulated in NIS, and *vice versa* ¹, supporting this possibility. Single cell transcriptomic analyses might address this question in the future. Since the current work focuses on chromatin structural alterations rather than transcriptional changes, we show these data (Reviewers Fig. R1) only to the reviewers and editors.

Considering the technical quality of the work, the interest for a wide audience and the important scientific advances presented in the manuscript, it very well suited for publication in Nature Communications.

Reviewer #2:

Parry et al. present a manuscript which is a natural extension of their recent publication in Nature Cell Biology (Hoare et al., 2016) in which ‘lateral induction’ of senescence is proposed as a novel route to senescence. The authors employ the following models of senescence: RAS-induced senescence (RIS; IMR90 ER:RAS) and Notch-induced senescence (NIS). RIS is associated with the induction of senescence-associated heterochromatic foci (SAHFs) and an increase in nucleosome-free regions (NFRs; ATAC-Seq). By contrast, NIS occurs in the absence of SAHFs, and can act to suppress SAHFs and NFRs in a RIS context. The authors propose that non-autonomous NOTCH 1 signaling and JAG1 (JAGGED1, Notch ligand) expression repress SAHFs in adjacent cells, in part by repression of the chromatin architecture proteins HMGA1 and 2. The authors perform ATAC-sequencing, ChIP-seq for the enhancer-associated modification H3K27Ac and GO analysis for RIS, NIS, N+RIS and growing IMR90 cells to identify NFRs. This is combined with ectopic N1ICD and HMGA1 knockdown to assess the influence on RIS-specific NFRs. The authors then propose that tumour cells ‘repress’ SAHFs and RIS-specific NFRs in a cell-contact dependent manner through the use of three cancer cell lines selected for low, medium and high JAG1 expression. Finally, data from TCGA and KMPlot is presented which indicates an anti-correlation between HEY-L (canonical NOTCH1 target gene) and HMGA1 in a panel of tumour types, and that HEY-L-Low/HMGA1-High expression shows increased survival in lung-squamous cell carcinoma. Taken together, this work highlights a role for HMGA1 in RIS-specific NFR formation.

The manuscript was a pleasure to read, is well written with a good logic flow and generally sound statements and conclusions. The discovery of differences between RIS and NIS NFRs is novel, and considerable effort has been taken to explore the JAG1-HMGA1 axis at a global level. Detailed below are some issues and requests for additional details which would help improve the manuscript further.

Major Issues

1. Figure 6 and Supplementary Figure 6.

MCF7, A549 and Hep3B cells are selected based on their JAG1 protein expression (low, medium and high, respectively, Figure 6a). Co-culture of IMR90 ER:RAS results in a decrease in the percentage of SAHF positive nuclei which is inversely related to the JAG1 expression (Figure 6b), and similar results are presented for HMGA1 and HMGA2 mRNA levels (Figure 6d). However, there seems to be a disconnect between JAG1 levels and the impact in co-culture experiments on HEY-L and HES1 levels (Supplementary Figure 6c), i.e. they do not directly relate to JAG1 levels, which suggests that the effects that are being attributed to JAG1 may be an oversimplification. This internal inconsistency is skirted over within the text and should be addressed more specifically.

The relationship between transcription input and output is not necessarily linear but it could involve thresholds and/or saturation points ⁴. Although HEYL, HEY1, and HES1 are known as ‘canonical targets’ of NOTCH, their transcriptional regulation by NOTCH signalling is highly complex: for example, unique combinations or modes of interaction between the 5 ligands and 4 NOTCH receptors can provide preferential induction of certain targets ⁵. Furthermore, these NOTCH targets are also regulated by other transcription factors. As ‘signal sending cells’, we use different tumor cell lines, which might differentially express other NOTCH ligands or other NOTCH modulators, conferring additional complexity. Thus, we do not exclude the possibility that, in addition to JAG1 levels, additional factors might be involved in the phenotype observed in the co-culture system. As a means of highlighting this, we have altered the text (page 17, line 9 onwards). Nevertheless, the combination of data previously presented in this manuscript together with new experiments (suggested by the reviewer, see major comment 2 below) strongly support our major conclusions.

2. The authors perform transwell experiments with MCF7/A549/Hep3B cells and IMR90 ER:RAS which indicate that co-culture is important for repression of SAHF formation (Supplemental Figure 6b) and co-culture in the presence of DAPT (γ -secretase inhibitor which inhibits downstream signaling of NIS) (Figure 6b). However, given that these three cancer cell lines are likely to be different many other potential cell surface proteins in addition to JAG1, the authors should attempt a co-culture experiment combined with specific JAG1 knockdown and determine the impact on SAHF positive cells to establish that this is a JAG1-mediated effect. It may be that the

authors have attempted such a set of experiments and observed reduced viability with JAG1. If so, it would be helpful to see such data. Alternatively, inducible (and therefore titratable) expression of JAG1 could be an alternative route to assess this in an isogenic manner.

The reviewer makes an important point and we agree that demonstrating the specific effect of JAG1 is essential. To achieve this, we used CRISPR-Cas9 technology to generate A549 cells lacking endogenous JAG1. Two knockout clones were isolated; clone 1 had a 5bp deletion in the first allele and a 1bp deletion in the second allele whilst clone 2 had a 1bp deletion in the first allele and a 14bp deletion in the second allele (new Supplemental Fig. 7d). Different guide sequences were used for the two clones although both targeted the second exon. A control was generated by transfecting cells with Cas9 but omitting any guide-RNA (-gRNA control cells). By flow cytometry (using a JAG1/2 antibody) we found that both clones 1 and 2 had reduced total levels of cell-surface JAG1 and JAG2 ligand (new Supplementary Fig. 7e). Moreover, JAG1 protein was not detectable by immunoblotting of knockout clones using a JAG1 specific antibody (new Fig. 6f). Proliferation was not altered in JAG1 knockout cells relative to -gRNA control and parental cells, at least not in a consistent manner (clone 1 had a slightly increased proliferation rate whilst clone 2 had a slightly reduced rate) (new Supplementary Fig. 7f). Co-culture of JAG1 knockout A549 cells with red IMR90 ER:RAS^{G12V} cells in the presence of 4OHT had little effect on SAHF formation in red cells (new Fig. 6g). Consistent with our previous results, co-culture with -gRNA control and parental cells significantly reduced the number of SAHF positive cells within the population (new Fig. 6g). Together, these data indicate that JAG1 is critical for SAHF repression by A549 cells in adjacent (senescent) cells.

In addition to JAG1 knockout A549 cells, we also generated MCF7 cells containing doxycycline (DOX)-inducible JAG1 fused to mVenus (JAG1-mVenus). By immunoblotting, we observed low-level expression of JAG1-mVenus even in the absence of DOX, likely caused by 'leaky' transcription of the construct (new Supplemental Fig. 7g). Addition of 10 ng/mL of DOX to the culture was sufficient to induce JAG1 to comparable levels as those observed endogenously in Hep3B cells (new Supplemental Fig. 7g). Co-culture of MCF7 cells containing inducible JAG1 with red IMR90 ER:RAS^{G12V} cells was sufficient to reduce the number of SAHF positive red cells even in the absence of DOX (reflecting the slightly increased levels of JAG1 expressed) and completely repress SAHF formation in red cells in the presence of DOX (Supplemental Fig. 7h). We believe that this system demonstrates an anti-correlation between JAG1 levels on 'signal-sending cells' and the number of SAHF positive fibroblasts. We acknowledge that we still cannot exclude the effects of other cell-contact mediated signaling pathways (page 18, line 16), although our data showing that DAPT treatment can rescue SAHFs in co-culture experiments (Fig. 6b) suggests that the effect is mediated through at least one of the NOTCH receptors.

Minor Issues

1. The work presented within Figure 1 raises the question as to whether ectopic expression of N1ICD in established RIS can alter SAHF formation once they have been established, and is so, shift the senescence state from RIS to NIS. In other words, is there plasticity within the RIS phenotype (SAHFs and NFRs) or is this phenotype 'locked in' once SAHF are established.

This is an interesting question and indeed previous work ⁶ has demonstrated that depletion of HMGAs can abrogate SAHFs even after they have been established. To determine whether SAHF formation is reversible by N1ICD, we infected IMR90 cells with constitutive oncogenic HRAS and DOX inducible N1ICD. Following the establishment of senescence (day 8 post infection), 1000 ng/mL of DOX was added for 3 days before the percentage of SAHF positive cells was determined (day 11 post infection). The addition of DOX (and induction of N1ICD) significantly reduced the number of SAHF positive cells relative to populations that were not treated, suggesting that NOTCH signalling can reverse at least SAHF formation even after establishment (new Supplemental Fig. 1g). The relationship between SAHFs and nucleosome positioning is an extremely interesting question and we are hoping to answer this in the future: the dynamic plasticity of nucleosome positioning in the context of high-order chromatin structure.

2. Page 11 Line 20 onwards. Can the authors provide the details for how many peaks scored for 3/3 samples for each of the NFRs.

As suggested by reviewer 3 we have now adopted a more quantitative approach to the analysis of our ATAC-seq data. Instead of relying on peak calling, we have used differential binding analysis (a combination of THOR and edgeR) to detect regions where chromatin accessibility is significantly altered in RIS and NIS cells relative to growing cells. Whilst this information can be provided if required, it no-longer impacts downstream analysis or interpretation of the data.

3. Page 12 Line 20 onwards. Is the data presented for intersected consensus peaks for 2 out of 3 samples again?

In our initial submission, all presented data were consensus peaks present in at least 2 samples. As detailed above, these have now been replaced with quantitative differential binding analysis.

4. Page 13 Line 5 onwards. NFRs within 300bp of a gene TSS were taken forward for GO analyses and the authors describe common pathways with previous RNA-seq data. Can they provide details of the specific genes which were identified by this analysis and of these which are specifically in common with the RNA-seq data. i.e. do

the genes which alter in the RNA-seq data also have altered NFRs, or it a 'coincidental' overlap in pathways which is being reported.

On average, genes that become more accessible (opened) in RIS cells relative to growing cells as indicated by ATAC-seq are also transcriptionally upregulated in RIS cells relative to growing cells as indicated by mRNA-seq (new Fig. 4f, top panel). The same is true of the NIS condition where, on average, genes that are opened in NIS cells relative to growing cells are also upregulated in NIS cells relative to growing cells (new Fig. 4f, bottom panel).

To determine the exact overlaps between these gene sets, we intersected the lists of genes that are significantly upregulated in RIS or NIS (determined RNA-seq, adjusted p-value <0.05) with the lists of genes that become more accessible (opened) in RIS and NIS cells (relative to growing cells, regions within ± 500 bp of the TSS). Of the genes opened in RIS cells 110 of 463 (23.7%) were also significantly upregulated at the mRNA level in RIS cells (Reviewers Fig. R2a). Of the genes opened in NIS cells 225 of 582 (38.7%) were also significantly upregulated in NIS cells (Reviewers Fig. R2b). To investigate whether the overlap between GO pathways in our paper (using ATAC-seq) and previous studies (using RNA-seq) are coincidental, we determined the intersection between genes that are both upregulated and opened in the RIS condition with genes that are both opened and present in the GO term 'inflammatory response' (the term most enriched in our pathway analysis, new Fig. 4g). Eleven genes were upregulated, opened and present within this GO term (Reviewers Fig. R2c). Together our data show that, whilst not all genes that become significantly more accessible (in RIS or NIS relative to growing cells) are also significantly upregulated, there is a substantial correlation between increased chromatin accessibility and gene transcription.

Regions that become significantly more/less accessible in RIS and NIS cells relative to growing cells were annotated using the computational package 'Homer'. These annotations provide the co-ordinates for each accessible region, their distance to the nearest TSS and the name of the closest gene. As a resource these annotations have been provided in Supplementary Table 1.

5. Page 13 line 16. Since this manuscript was submitted there have been further publications which explore the role of pro/anti-apoptotic pathways. Perhaps the authors could extend their bibliography to include these papers.

As suggested by reviewer 3, we have now adopted a more quantitative approach to the analysis of our ATAC-seq data. Instead of relying on peak calling, we have instead used differential binding analysis (a combination of THOR and edgeR) to detect regions where chromatin accessibility is significantly altered in RIS and NIS relative to growing cells. Whilst the overall message of our manuscript remains the same, we no longer observe a substantial overlap of regions that become more accessible in both

RIS cells and NIS cells relative to growing cells (3,073 such regions were identified, previously called ‘novel shared’ regions). As a result, very few genes become more accessible in both conditions and these are not enriched in any particular GO-term. Since we can no longer detect significantly increased chromatin accessibility at the promoters of pro/anti-apoptotic genes in RIS and NIS cells, this discussion has been removed from the revised manuscript.

6. Page 14 Line 18 onwards. Was ATAC-seq performed for both IMR90 ER:RAS and IMR90 ER:RAS+shHMGA1 or just the later. If ATAC-seq was performed for both samples have does this data compare to the ATAC-seq data presented earlier in the manuscript?

In this particular experiment, we have added ‘IMR90 ER:RAS+shHMGA1’ samples to the initial datasets. Based on the reviewer 3’s suggestion, we have included all the datasets for PCA and two independent quantitative normalization approaches, which agree with each other, largely, if not completely, in order to eliminate batch effects. In addition to the datasets presented in this manuscript, we have generated ATAC-seq libraries from growing and senescent IMR90 cells on many different occasions. In all cases, our data reproducibly support the conclusions presented earlier in the manuscript; we observe increased chromatin accessibility in RIS cells relative to growing cells, especially in gene distal regions of the genome.

7. Page 15 Line 20 onwards. Did the authors perform ATAC-seq for NIS+shHMGA1, and is so what did this data support their assertion that NIS occurs via an HMGA-independent mechanism.

We did not perform ATAC-seq in this context. In NIS, the levels of HMGA1 are already very low at both the protein and mRNA level (indeed, the levels are even lower than we can achieve using a short hairpin against HMGA1; Fig. 3a). Since HMGA1 knockdown in the context of RIS does not induce the formation of NIS specific open regions (new Fig. 5c, f), we conclude that the mechanism for the formation of these is HMGA1 independent.

8. Page 17 Line 7 onwards. The authors ask whether tumour cell lines could repress RIS-unique NFR formation in IMR90 ER:RAS. What is the overlap with the ATAC-seq performed in this paragraph with NIS NFRs and N+RIS NFRs? How does this compare with the intersections for the cancer cell lines with increasing JAG1 expression.

As part of the analyses requested by reviewer 3, we have now performed differential binding analysis of RIS cells co-cultured with tumour cells (RIS+MCF7, RIS+A549 and RIS+Hep3B) relative to RIS cells cultured alone (new Fig. 6h, i). Using the metrics computed by edgeR, we generated volcano plots showing the log₂ fold-change (of accessibility) and $-\log_{10}$ FDR for each of the regions. We colored the data points to indicate whether they also become significantly more accessible in RIS vs growing (opened in RIS, similar to ‘RIS specific NFRs’) or NIS vs growing (opened in NIS,

similar to 'NIS specific NFRs'). Co-culture with tumour cell lines was sufficient to increase the accessibility of regions that were also opened in NIS relative to growing cells. Co-culture with MCF7, A549 or Hep3B cells significantly increased the accessibility of 6,948, 10,303 and 14,064 sites respectively. These data correlate with the levels of JAG1 expressed by each cell line and indicate that the tumour cell lines can both repress 'RIS specific chromatin opening' and promote 'NIS specific chromatin opening'. As with SAHF formation, co-culture with MCF7 cells had an effect on chromatin accessibility (albeit less of an effect than A549 and Hep3B cells). The basal effect of MCF7 cells could be caused by low levels of JAG1 (not detected by western blotting) or by other NOTCH1 ligands.

9. Page 18 Line 10 onwards. I think that microarray data for just 3,554 breast cancer patients is presented within the figure, rather than the full 5,143 as stated. Likewise, I think data for just 1,145 lung cancer patients is shown. Please provide details as to whether is RFS, OS etc.

We thank the reviewers for spotting this error. Graphs labels have also been updated to show that the measurement is overall survival. Note that we have replaced the analysis of lung and breast cancers for *HMGA1* and *HEYL* with a subtype specific analysis of lung cancer alone (adenocarcinoma and squamous cell carcinoma). We chose to do this because we noticed that high *HEYL* expression appears to be beneficial in lung adenocarcinoma (new Supplementary Fig. 9d) but detrimental in lung squamous cell carcinoma (new Fig. 7c).

10. Page 18 Line 4. Should this read ' $r=-0.4842$ ' rather than ' $r=0.4842$ '.

We thank the reviewers for spotting this error - it has been corrected.

11. Page 22 Line 4 onwards. Please provide the dose(s) of each compound used within this manuscript.

For clarity, doses have been added to the methods section, the figure legends and the cartoon figures that are used to illustrate experimental design (e.g. new Fig. 1b, new Fig. 2a, d).

12. Page 24 Line 2. Please provide detailed information as to how a SAHF and SADS positive cell (nuclei?) were defined as positive versus negative. In some of the images it looks as if there may still be a small number of SAHFs present, and it is unclear how this criteria has been applied.

SAHF were defined as previously⁶⁻⁹ by staining permeabilized nuclei using DAPI and scoring each nucleus within the population as positive or negative. At least 200 cells from random fields around the coverslip were scored for each biological replicate. Although we understand that such analyses can be subjective, we generally find that it is quite simple to determine whether a nucleus is SAHF positive or negative when DAPI staining is observed under the microscope. Specifically, a SAHF positive

nucleus will have multiple DAPI dense foci (>5) surrounded by DAPI poor regions. Indeed, in our laboratory all researchers agree that approximately 60% of our RIS cells are 'SAHF positive'. To reduce any potential bias and lessen error within this study, the cultures were independently counted by two researchers (AP and MH) and in all cases the total percentages were found to be similar

SADS positive cells are cells, in which the signal from DNA fluorescence in-situ hybridization (DNA-FISH) against α -satellite repeat sequences is noticeably distended. Within a single SADS positive nucleus, the majority of the signals are extended, making it quite easy to distinguish from SADS negative nuclei¹⁰. At least 200 cells per biological replicate from random locations around the coverslip were scored. Images of DNA-FISH stained nuclei showing SADS positive RIS cells and SADS negative growing cells have been added to new Supplementary Fig. 1a.

13. Page 25. Please provide details of validation of the anti-H3K27Ac and anti-H3K4me1 antibodies prior to ChIP-seq. Was any post ChIP-seq validation performed? These are purified monoclonal antibodies produced in-house by Prof. Hiroshi Kimura and colleagues. The same clones are also commercially available and are recommended for ChIP experiments by their supplier (H3K27Ac, Active Motif, Clone MABI0309, cat# 39685; H3K4me1, Active Motif, Clone MABI0302, cat# 30635). Both antibodies were validated for chromatin immunoprecipitation in the primary publication for which they were generated¹¹ and the clones have since been validated for ChIP assays in other publications¹²⁻¹⁴.

To ensure data quality, we use FastQC to check basic statistics and also MACS2 to find peaks with an FDR cutoff of 5%. A summary of the peaks detected in each dataset is presented below (G = Growing, S= RIS, number = biological replicate number), including the number with a fold enrichment above 5% (foldEnrichment_5) and the percentage of the reads that are in peaks (RiP%).

file_name	peak_number	foldEnrichment_5	RiP%
H3K4me1_G_1	267869	126784	48
H3K4me1_G_2	250341	123337	52
H3K4me1_G_3	266591	142013	63
H3K4me1_G_4	237172	111965	46
H3K4me1_S_1	318465	154440	54
H3K4me1_S_2	324244	142351	58
H3K4me1_S_3	304497	158409	62
H3K4me1_S_4	316276	161523	60

file_name	peak_number	foldEnrichment_5	RiP%
-----------	-------------	------------------	------

H3K27Ac_G_1	116454	66493	42
H3K27Ac_G_2	118339	65652	42
H3K27Ac_G_3	118791	67671	44
H3K27Ac_G_4	112836	64687	41
H3K27Ac_S_1	138394	83121	58
H3K27Ac_S_2	139599	82562	57
H3K27Ac_S_3	146259	80010	58
H3K27Ac_S_4	143240	82086	59

Please also see the point below.

14. Page 27 Line 17 onwards. “Enhancers were identified based on our own H3K4me1 and H3K27ac ...ChIP-seq datasets” – are these published, if so please provide a reference. Alternatively, how have these been validated to ensure that enhancers were accurately identified.

The ChIP-seq datasets have not been previously published and we agree that it is important to validate that our ‘enhancers’ were accurately annotated. In order to achieve this, we downloaded enhancer annotations from two repositories: ROADMAP and Broad ChromHMM. Approximately 88% of the regions we identify as enhancers using our own ChIP-seq datasets (H3K4me1 and H3K27ac) are also present in at least one of these publically available annotations (Reviewers Fig. R3), giving us high confidence in our own annotations.

15. Figure 1. Please provide more technical details of how the experiment was performed, i.e. at what time point is OHT added etc.

A figure providing details on experimental design has now been included. Constitutive N1ICD-Flag is introduced by retroviral infection and cells are selected using 75 μ g/mL hygromycin-B for 3 days. After 3-5 days of recovery 100nM 4OHT is added for a further 6 days to induce the expression of HRAS^{G12V}.

16. Figure 1b. Is the image of the RAS nucleus representative?

This is a typical SAHF positive nucleus. As shown in Fig. 1c and our previous studies¹⁵, such SAHF positive cells are ~60% of the population in the ER:RAS system. This particular nucleus was chosen for the figure to illustrate the striking difference between RIS cells (that often contain these structures) and NIS cells (which never do: N1ICD and RAS/N1ICD nuclei are also representative). To clarify this, we have added % of SAHF-positive cells in each setting (new Fig. 1b).

17. Figure 1e. As per notes above, please provide details of the DAPT dose employed. The level of SAHFs at time zero for the mVenus-dnMAML1 appears to be significantly

(**) elevated relative to the mVenus control. Please provide an explanation for this. Please also provide a representative image of the mVenus + DAPT.

There are a small number of basal (pre)senescent cells in normal 'growing' populations due to culture/replicative stress. In this experiment, DAPT was added at the same time as 4OHT (to induce RAS), thus at day 0, mVenus and mVenus+DAPT cells are indeed the same condition. In contrast, dnMAML1 was stably introduced before the experiment, thus at day 0, any basal NOTCH signaling was already repressed in mVenus+dnMAML1 cells. This could 'boost' the SAHF phenotype at day 0. As a comparison, we have added representative images of mVenus(+RAS), mVenus+DAPT(+RAS) at day 7. Consistent with this, treatment of cells with 30 μ M DAPT in the absence of 4OHT for 6 days is also sufficient to 'boost' the SAHF phenotype in a subset of cells within the population (new Supplemental Fig. 1e).

The reviewer has asked to 'provide a representative image of the mVenus + DAPT'. We assumed that he/she means mVenus+DAPT(+RAS), which we have added to new Fig. 1e. In case the reviewer means mVenus cells treated with DAPT alone, representative images and SAHF counts from IMR90 ER:RAS^{G12V} cells treated with 10 μ M DAPT alone for 6 days (in the absence of 4OHT to induce RAS) are provided in Reviewers Fig. R4. Like mVenus-dnMAML1, exposure to DAPT for 6 days is also sufficient to 'boost' the SAHF phenotype in a subset of cells within an otherwise growing population (without 4OHT treatment).

18. Figure 2e. From the images presented, it looks as if the RPE1 JAG1 mVenus nuclei (Blue) have are SAHF positive. Can RAS drive SAHF formation in adjacent RPE1 nuclei?

We find that, compared to human fibroblasts, RPE1 nuclear staining by DAPI tends to be more heterogeneous, often exhibiting a few DAPI-dense foci in the normal 'growing' condition. These DAPI-dense foci (small or large depending on the cell type) are common in mammalian cells, but are not SAHFs, which are formed by spatial aggregation of pre-existing heterochromatic domains following the establishment of senescence. Note, unlike SAHF-positive nuclei, regions outside of these common DAPI-dense foci are normally also stained by DAPI (another well-known example is in MEFs, which show multiple DAPI-dense constitutive heterochromatin foci on top of normal DAPI-staining pattern in a proliferative state). This is now clarified in the legend.

19. Figure 5c. Do the width of NFRs alter in RIS+shHMGA1 relative to RIS, especially in cluster 1. If so, what is the authors interpretation of this data?

We can see the effect that the reviewer is referring to but believe that this is a visual error introduced during generation of the heat maps or the PDF file. Much of the data has been reanalyzed using differential binding analyses, as requested by reviewer 3. As a result, this figure has been regenerated and we can no longer see such an effect.

When viewed using a genome browser, accessible regions in the RIS+shHMGA1 condition are of comparable width to those in RIS cells (new Supplemental Fig. 6 a-d).

20. Figure 6b. How does this data relate to the data in Supplemental Figure S1E? What does of DAPT was used? Please provide the Alone +DAPT data for direct comparison and relate this to the findings with the co-culture with the cancer cell lines. In Supplemental Fig. 1e, we titrate the dose of DAPT to determine if this has an effect on SAHF formation in IMR90 ER:HRAS^{G12V} cells. We find that 10 μ M of DAPT in addition to 4OHT increases the number of SAHF positive cells within the population from ~60% to ~80% (a significant increase). In Fig. 6b, 10 μ M of DAPT is used in co-cultures to rescue SAHF formation and again ~80% of the fibroblasts form SAHF in the presence of 10 μ M DAPT even when co-cultured with tumour cell lines. Therefore, these data reproduce our findings in Supplemental Fig. 1e but in the context of a co-culture. As requested, data for the 'Alone +DAPT' condition have now been provided for clarity (new Fig. 6c).

21. Supplemental Figure 4b. The PCA analysis indicates that NIS-1 and N+RIS-1 do not cluster with their corresponding -2 and -3 samples. Were both of these excluded from future analysis? See also text on page 11 Line 15.

NIS-1 and N+RIS-1 were excluded from downstream analysis based on both their clustering behavior and the low percentage of reads in peaks (RiP%).

Figure legends

Please add details of compound doses used within a given experiment for clarity.

Doses have now been added to each of the figure legends and to the cartoon diagrams depicting experimental designs.

Consistency.

Please check the manuscript for general consistency, e.g. IMR90 ER:RAS or IMR90 ER:HRAS, HEYL or HEY-L.

We thank the reviewer for pointing this out – the manuscript has now been checked for consistency.

Reviewer #3:

In this paper, Parry and colleagues report some mechanistic observations differentiating NOTCH-induced from RAS-induced senescence, especially on how juxtacrine signaling might induce a chain of events leading to large chromatin re-organization. They also show the level of expression of some of these genes to be predictive of survival of lung and breast cancer patients.

I have very limited knowledge of cellular senescence and this field in general, so I am not able to give a proper feedback about the novelty of the manuscript. Overall, I found it easy to follow (except for a number of acronyms and use of jargon) and providing interesting observations whose relevance for primary tumors is computationally validated using publicly available TCGA datasets. Nevertheless, I think the analyses of the generated genome-wide profiles of chromatin accessibility are affected by some major flaws that currently mine one of the main conclusions of the paper, namely chromatin remodeling at tens of thousands of loci in senescent cells. These shortcomings must be addressed in order to justify publication of the manuscript in Nature Communications.

The authors made a genuine effort in generating multiple replicates of the ATAC-seq libraries per condition, but a larger analytical effort is required to support the authors' conclusions. The results of the functional analyses (e.g. Fig. 4F) are promising and supported by a few anecdotal screenshots provided in Fig. 4 and Supp. Fig. 5. Nevertheless, these need to be confirmed by more thorough analyses of the data (see major comments).

Major comments:

1. The major difference in the total number of accessible regions seen between the growing and the RIS/NIS/R+N samples (Fig, 4D) is currently not strongly supported by data analyses. While the results seem to point to a strong qualitative difference in the number of open-chromatin regions, the way the analyses were performed raises a number of questions on how this difference would be rather quantitative than qualitative.

1A. Most of the peaks might have not been identified in the growing samples given that only 1 out of 4 replicates for that condition showed a signal-to-noise ratio (considering the RIP% as a surrogate for that) in line with most of the samples in the other conditions (Supp. Fig. 4A). Given the authors required a peak to be present in at least two samples, separately for each condition, this clearly put the growing samples at a disadvantage in terms of sensitivity.

We agree that MACS peak calling has limitations and have now used differential binding analysis as suggested below (comment 1A, below). We did however note, by quantifying pairwise overlaps, that there was a good agreement between peaks detected across all 4 of the growing samples. For example, the majority of peaks (>70%) detected in replicate 1 were also present in replicates 2, 3 and 4 (Reviewers Fig. R5a).

1B. As a quality control, the authors should run MACS using increasingly more lenient thresholds. If the number of peaks called between the growing and the other conditions

converges to more similar numbers, this would be a good indication that the difference is mainly driven by the signal-to-noise issue highlighted above.

As that we have now used differential binding analysis to quantify the differences in accessibility, suggested in comment 1C, we have not found it necessary to run MACS at additional thresholds. Of course, this can be performed if required.

1C. In any case, the authors should use a more appropriate approach to quantify the differences in accessibility between conditions. A standard way of doing this is to apply approaches such as edgeR (PMID: 19910308) or DEseq (PMID: 20979621) starting from the normalized counts of the identified peaks. These methods have been conveniently wrapped into a Bioconductor package called DiffBind.

As suggested, we have now used differential binding analyses to quantify differences in accessibility between conditions. In order to confidently identify regions that become more/less accessible in RIS and NIS cells relative to growing cells, we used both THOR (Alhoff *et al.*, 2016) and edgeR (Robinson *et al.*, 2010) before taking the intersect of the two outputs (see methods page 30, line 17 onwards). By taking the intersect, we eliminated many regions that appeared, at least in the genome browser, to be false positives (Reviewers Fig. R5b). This stringent approach gave us a good 'base-set' of regions that are altered in these two distinct types of senescence (RIS and NIS) for downstream analyses (new Fig. 4c). As with the peak calling approach previously reported, we observed that many regions become more accessible in RIS relative to growing cells whilst few regions become less accessible. Unlike the data generated using MACS, we now also detected many regions that become significantly less accessible in NIS cells relative to growing cells (new Fig. 4c). Moreover, most of the regions that became more accessible in RIS and NIS were unique to each phenotype, with very few of the 'novel shared' regions detected before (new Fig. 4d). As with our previous analysis, most of the novel accessible regions in RIS or NIS were in gene distal locations (new Fig. 4e), but those that were close to transcriptional start sites (TSSs) correlated well with the expression of the associated genes on the mRNA-level (as detected using mRNA-seq, new Fig. 4f). Genes that become more accessible in RIS were highly enriched in gene ontology terms such as 'inflammatory response' and 'cytokine secretion' whilst genes that become less accessible in RIS were associated with the regulation of the cell cycle, reflecting cell cycle arrest. This new analysis does not change the overall message of our paper, indeed most of the observations remain valid or have become more robust, thus increasing our confidence in the wide scale alterations to the nucleosome landscape observed in RIS and NIS. We appreciate the helpful suggestion by the reviewer.

For downstream analyses where the RIS phenotype has been manipulated, for example by co-expressing N1ICD or by co-culture with tumour cell lines, we chose to use edgeR (Fig. 5b, c; Fig 6g). edgeR and THOR largely agreed on regions that become more/ less accessible in these comparisons (Reviewers Fig. R5c) and the use

of one method allowed us to present intuitive volcano plots where points are coloured as either 'opened in RIS versus growing' or 'opened in NIS versus growing' (Fig. 5b, c; Fig. 6h). From these plots, one can quickly and easily interpret that: 1) NOTCH signalling repressed the formation of RIS-driven accessible regions (those that become more accessible in RIS cells versus growing cells) and induced the formation of NIS-driven accessible regions (new Fig. 5b); 2) HMGA1 depletion inhibited the formation of a subset of RIS-driven accessible regions but did not induce the formation of NIS-driven accessible regions (new Fig. 5c); 3) Co-culture with tumour cell lines repressed the formation of RIS-driven accessible regions and induced the formation of NIS-driven accessible regions (new Fig. 6h), and that the strength of this effect correlated with the levels of JAG1 expressed by the tumour cell line (consistent with their effects on SAHF formation).

1D. Given the limitations of the current analyses, the results shown in Supp. Fig. 4C-E cannot be fully evaluated. Besides, given the standard alignment pipeline only considers uniquely mappable reads, the repetitive element identified might actually be only the tip of the iceberg of a much larger phenomenon, which would require the analysis of multi-mapping reads (PMID: 22124482). This should be at least acknowledged if not investigated.

Our conclusion that distal regulatory elements and repeat elements become more accessible in RIS and NIS relative to growing cells has not been altered by the use of THOR and edgeR to identify accessible regions, indeed the effect appears to be even more pronounced when only regions that become more accessible are considered and compared to regions that are unchanged (new Fig. 4e; new Supplemental Fig. 5b, c, d). In particular, regions that become more accessible in RIS cells appear to be enriched in certain classes of repeat element, namely DNA repeats, LINEs, SINEs and LTRs. These data are consistent with a previous publication that presents FAIRE-seq data generated from cells undergoing replicative senescence ¹⁶. Because our manuscript largely asks how NOTCH signaling effects overall chromatin structure (in RIS and NIS cells), rather than how repeat elements are altered *per se*, we have not changed our analysis to include multi-mapping reads. As suggested, we have however acknowledged in the text that our analysis excludes multi-mapping reads (page 12, line 11 onwards) and therefore does not necessarily capture all of the alterations within these regions. We would be excited to study repetitive elements in RIS and NIS in more depth in the future.

2. A description on how the PCA showed in Supp. Fig. 4B was performed is lacking. The first principal component seems to separate the samples by RIP%. This is in fact showing that there is more variability between NIS replicate 1 and N+R replicate 1 and all the other replicates in these two conditions, rather than compared to the growing samples. This would suggest excluding these samples from any downstream analyses,

unless the PCA was performed using the significance level of peaks rather than the normalized signals (if this is the case, the PCA should be repeated).

A description on how the PCA shown in Supplemental Fig. 4b has now been included in the methods (page 34, line 1 onwards). As suggested, the PCA analyses were performed using normalized signals from each of the samples rather than the significance level of peaks. NIS replicate 1 and N+RIS replicate 1 were excluded from downstream analyses based on their low RiP%. The PCA presented in Supplemental Fig. 4B has been replaced and now includes all of the samples used for downstream analysis, including ATAC-seq samples generated from co-cultures and with the short-hairpin targeting HMGA1. It seems that the first principle component no longer separates samples based on RiP%, as samples with very similar values (e.g. RIS+MCF7 and RIS) do not cluster along this axis whilst some samples with very different values cluster together.

3. Similarly, a description on how the clustering in Fig. 4C was carried out is lacking and should be provided.

A description on how the clustering analysis (new Fig. 5a) has now been included in the methods section (page 34, line 1 onwards).

4. The “RIS+shHMGA1” ATAC-seq sample should also be included in the PCA analysis. It should be quality controlled (e.g. what is the RIP for this sample?) and processed as suggested above for the other samples (e.g. by directly comparing the signals from the RIS+shHMGA1 condition with the RIS, Fig. 5A). The same considerations apply to the samples shown in Fig. 6.

The RIS+shHMGA1 and RIS+ tumour cell line samples have now been included in Supplemental Fig. 4a, b. The RiP% of these were found to be in line with the values of other samples and the replicates clustered well by PCA analysis.

5. The conclusion that “HMGA1 is a key regulator of chromatin structure at the nucleosome level” (page 15) is currently not fully supported by data analysis. Given binding-motifs for HMGA1 have been published (for example see the collection HOCOMOCO: <http://hocomoco11.autosome.ru/>), the authors should at least perform an over-representation analysis for known transcription-factor-binding-motifs in the different groups identified in Figure 5. This unbiased analysis might confirm that the HMGA1 motif is indeed highly enriched in the RIS-unique group and importantly, might point to other transcription factors that could differentially bind these regions.

Whilst binding-motifs for HMGA1 are published, to our knowledge the supporting evidence for these is weak. The majority of studies suggest that HMGA1 binds non-specifically to AT-rich sequences within the minor groove of DNA ¹⁷. Moreover, a recent study that mapped the distribution of HMGA1 genome wide concluded that binding is dominated by AT-content alone rather than by any specific motif and that there is little focal enrichment of HMGA1 across the genome ¹⁸.

In line with the view that HMGA1 binding is non-specific, the motif published in HOCOMOCO (http://hocomoco11.autosome.ru/motif/HMGA1_HUMAN.H11MO.0.D) has a low quality score (categorized as 'D'), which '*provides only a rough description of a binding pattern and should be used primarily in exploration studies*' (HOCOMOCO website). Nevertheless, we were able to detect highly significant enrichment of the HMGA1 motif around regions of accessible chromatin increased in the RIS condition ('opened in RIS', Reviewers Fig. R6b), and indeed this enrichment was much greater than in regions of accessible chromatin unique to the NIS condition ('opened in NIS'). The HMGA1 motif was found to be enriched in the regions flanking the centre of the ATAC-seq peaks (Reviewers Fig. R6b), rather than the 'typical' profile that might be expected of a transcription factor that binds directly to a nucleosome free region (as for the motifs in new Supplemental Fig. 5e, f). The enrichment of the HMGA1 motif in the regions flanking RAS-driven accessible chromatin could indicate that HMGAs, together with other factors, facilitate the remodeling of nucleosomes around these regions, as was previously shown for the promoter of the *interleukin-2 receptor alpha* gene where HMGA1 is required to displace a nucleosome from a transcription factor binding site¹⁹. Whilst these data support our conclusions we have not included the results in the final manuscript as we are not confident that the motif is valid for this analysis, as discussed above.

As suggested in our discussion (page 21, lines 14 - 17), HMGA could facilitate the formation of RIS-driven accessible regions by increasing the accessibility of chromatin to other transcription factors, which would likely be essential for the establishment of the stable nucleosome-free regions detected here. As the reviewer suggests, unbiased *de novo* motif analysis could identify these other important factors. Using edgeR and THOR we find that most of the regions that become more accessible in RIS cells relative to growing cells are 'RIS-unique' (new Fig. 4c, d), thus we performed motif analyses on all of these (referred to in the updated manuscript as 'opened in RIS' or 'RIS-driven accessible regions') as well as the regions that become more accessible in NIS cells relative to growing cells (referred to as 'opened in NIS' or 'NIS-driven accessible regions'). We find that regions opened in RIS cells are highly enriched with common transcription factor motifs such as the bZIP motif bound by FOSL1 and others (new Supplemental Fig. 5e). The C/EBP β binding site was also highly enriched, consistent with the important role that C/EBP β plays in driving the expression of inflammatory SASP components²⁰. Other transcription factor binding motifs identified include the binding sites of NF2L1 and RORG, which are transcription factors with canonical roles in blood cells. Regions opened in NIS were also enriched with bZIP like motifs as well as for the RPB-J binding site (new Supplemental Fig. 5f), a critical DNA-binding factor of NOTCH that is required by N1ICD for chromatin binding²¹. We thank the reviewer for the helpful suggestion as these new analyses confirm the quality of our ATAC-seq data (because we detect strong enrichment of two transcription factor

motifs that might be expected: C/EBP β and RPB-J) and open up new avenues for future work (for example: are NF2L1 and RORG important for RAS-induced senescence?).

6. While the analysis in Fig. 5C is in principle well-done, there seems to be an issue with visualization (it might simply be a problem with the pdf). Some of the boxes are completely white, and in some of them a small version of the corresponding heatmap seems to appear in the bottom-left corner.

We believe this may have been a problem with the pdf file and apologise if the reviewer was unable to properly evaluate the data. We cannot see this effect in our own version of the pdf file but we have nevertheless regenerated the figure, hopefully solving the problem (new Fig. 5f).

7. One of the major points raised in the first part of the paper is the massive downregulation of HMGA genes in NOTCH-induced senescence (Fig. 3). The authors should look at the genomic landscapes of HMGA1 and HMGA2 in the ATAC-seq profiles they generated. This would at least provide further support for their conclusions (are the promoter and enhancers around these genes showing decreased accessibility?), if not to more a precise mechanistic hypothesis.

As suggested, we have now interrogated the genomic landscapes of HMGA1 and HMGA2 in the ATAC-seq profiles generated as well as the H3K27Ac ChIP-seq profiles generated (note that the latter were only generated in growing and RIS cells). We found that, in RIS cells, there is activation and increased accessibility at a number of enhancer elements (marked by H3K27Ac and H3K4me1) upstream of the HMGA1 gene, whereas the accessibility was not substantially altered in the promoter region (new Supplemental Fig. 6a). This increased accessibility at enhancers is repressed by NOTCH1 (N+RIS), suggesting that NOTCH could repress HMGA1 transcript at least in part by modulating the enhancer landscape. Although H3K27Ac levels were also highly increased across the HMGA2 promoter in RIS cells relative to growing cells (correlating with increased transcription), the accessibility of enhancer elements around the HMGA2 gene was not altered, at least within the range examined. Moreover, no significant alterations to ATAC-seq profiles were detected at the HMGA2 promoter in RIS or any other conditions (new Supplemental Fig. 6b). Note, we have previously shown that HMGA1 has a stronger impact on SAHF formation than HMGA2⁶. Since the increased accessibility at the HMGA1 enhancers is not fully repressed by sh-HMGA1 (RIS+shHMGA1), the data suggest an interesting possibility that 'HMGA1-independent' chromatin modulation by NOTCH might play a role in the initial repression of HMGA1, leading to the HMGA1-dependent effect.

Minor comments:

1. The authors show that both HEY1 and HEYL are significantly (although slightly)

anti-correlated with HMGA1 in terms of expression in cancer (Supp. Fig. 7). Despite that, they pick only HEYL because it is the most up-regulated in the experiments shown in Figs. 3 and Figs. 6. Nevertheless, HEY1 is only slightly less up-regulated than HEYL (Fig. 3F, but in a more robust way than HEYL) or not shown at all (Fig. 6). I think the authors should at least comment on (if not show) the results of running similar analyses to those shown in Fig. 7 and Supp. Fig. 8 for HEY1.

We acknowledge that HEY1 is also anti-correlated with HMGA and as suggested we have included HEY1 in the analysis originally presented in Supplemental Fig. 7 (new Supplemental Fig. 9) and replicated the analysis in Fig. 7 for HEY1 (presented in new Supplemental Fig. 10). HEY1 expression was prognostic of good outcome in lung cancer (new Supplemental Fig. 9d, e). Like HEYL, we see a significant negative correlation between HMGA1 and HEY1 in a wide variety of tumour types, although intriguingly these were not always the same tumours (new Supplemental Fig. 10a). Most notably, HEY1 and HMGA1 do not anti-correlate in lung squamous cell carcinoma and, unlike for HEYL, the relative expression levels of these two genes are not prognostic in this setting (new Supplemental Fig. 10c). However, there was a strong anti-correlation between HEY1 and HMGA1 in Kidney Renal Clear Cell cancer, which was prognostic of outcome (new Supplemental Fig. 10d, e). Whilst HEYL and HEY1 are both classed as canonical NOTCH1 target genes, our data shows that they do not necessarily behave in an identical manner *in vitro* or *in vivo*, and that the regulatory mechanisms underlying the functional outputs of NOTCH signaling are likely extremely complex.

Also note that, although our original km-plotter analysis used total lung cancer patients (original Supplemental Fig. 8a), since the TCGA data analysis (new Fig. 7) is focused on cancer subtype specific analysis, we decided to interrogate lung squamous cell (SCC) and lung adenocarcinoma separately using km-plotter. Interestingly, this analysis showed that high *HMGA1* expression predicts poor prognosis in adenocarcinoma but not in lung SCC (new Supplemental Fig. 9d, e). When TCGA data were analysed, '*HMGA1* high & *HEYL* low' status was predictive of good prognosis in lung SCC (new Fig. 7c). Thus, *HMGA1* can be predictive of either good prognosis (when combined with *HEYL*) or poor prognosis, depending on lung cancer subtype. For simplicity, we have removed the breast cancer data generated by km-plotter (original Supplemental Fig. 8b)

2. Page 11: "When interrogated using a genome browser, these were comparable to each other and to ENCODE DNase sequencing data from normal human lung fibroblasts (NHLF)". This is a good starting point but it has to be shown more formally, e.g. by doing a genome-wide correlation analysis between the ATAC-seq samples and a few published DNase profiles, including the NHLF and a few unrelated ones.

As suggested, we have systematically compared our own IMR90 datasets to publically available ATAC-seq and DNase-seq datasets from IMR90 cells and others. As

anticipated, our ATAC-seq samples clustered with publically available IMR90 ATAC-seq and DNase-seq datasets (new Supplemental Fig. 4c).

3. Fig. 4, the authors should provide scale for the y-axis when reporting screenshots from genome browsers (such as UCSC).

Scales have now been provided for all tracks taken from a genome browser.

4. In the methods section, page 27: “A similar filtering was carried out using Cutadapt as 5 described for RNA-seq data”. This step of the analysis is unclear and needs to be clarified.

As requested, this section of the methods has been expanded and clarified (page 30, lines 9-12).

5. In the abstract: NRFs = NFRs?

We thank the reviewer for spotting this error. Having now analysed our data using differential binding analysis it seems inappropriate to use the term ‘nucleosome free region’ to describe the regions being discussed, as some that are already accessible (technically ‘nucleosome free’) can become ‘more accessible’ or ‘less accessible’ in any of our comparisons. As such we have replaced the binary term ‘NFR’ with ‘more/less accessible’ or ‘opened/closed’.

6. The acronym SADS appears on page 6 for the first time and is never spelled out. SADS (senescence-associated distension of satellites) has now been spelled out. We thank the reviewer for spotting this error.

REFERENCES

1. Hoare, M. *et al.* NOTCH1 mediates a switch between two distinct secretomes during senescence. *Nat Cell Biol* **18**, 979–992 (2016).
2. Su, Q. *et al.* Jagged1 upregulation in prostate epithelial cells promotes formation of reactive stroma in the Pten null mouse model for prostate cancer. *Oncogene* **36**, 618–627 (2017).
3. Marthandan, S. *et al.* Similarities in Gene Expression Profiles during In Vitro Aging of Primary Human Embryonic Lung and Foreskin Fibroblasts. *Biomed Res Int* **2015**, 731938–17 (2015).
4. Nandagopal, N. *et al.* Dynamic Ligand Discrimination in the Notch Signaling Pathway. *Cell* **172**, 869–880.e19 (2018).
5. Bray, S. J. Notch signalling in context. *Nature Reviews Molecular Cell Biology* **17**, 722–735 (2016).
6. Narita, M. *et al.* A novel role for high-mobility group A proteins in cellular senescence and heterochromatin formation. *Cell* **126**, 503–514 (2006).
7. Narita, M. *et al.* Rb-mediated heterochromatin formation and silencing of E2F target genes during cellular senescence. *Cell* **113**, 703–716 (2003).

8. Chandra, T. *et al.* Independence of Repressive Histone Marks and Chromatin Compaction during Senescent Heterochromatic Layer Formation. *Mol. Cell* **47**, 203–214 (2012).
9. Sadaie, M. *et al.* Redistribution of the Lamin B1 genomic binding profile affects rearrangement of heterochromatic domains and SAHF formation during senescence. *Genes Dev.* **27**, 1800–1808 (2013).
10. Swanson, E. C., Manning, B., Zhang, H. & Lawrence, J. B. Higher-order unfolding of satellite heterochromatin is a consistent and early event in cell senescence. *J. Cell Biol.* **203**, 929–942 (2013).
11. Kimura, H., Hayashi-Takanaka, Y., Goto, Y., Takizawa, N. & Nozaki, N. The organization of histone H3 modifications as revealed by a panel of specific monoclonal antibodies. *Cell Struct. Funct.* **33**, 61–73 (2008).
12. Busby, M. *et al.* Systematic comparison of monoclonal versus polyclonal antibodies for mapping histone modifications by ChIP-seq. *Epigenetics & Chromatin* **9**, 49 (2016).
13. Semba, Y. *et al.* Chd2 regulates chromatin for proper gene expression toward differentiation in mouse embryonic stem cells. *Nucl. Acids Res.* **45**, 8758–8772 (2017).
14. Kotomura, N., Harada, N. & Ishihara, S. The Proportion of Chromatin Graded between Closed and Open States Determines the Level of Transcripts Derived from Distinct Promoters in the CYP19 Gene. *PLoS ONE* **10**, e0128282 (2015).
15. Young, A. R. J. *et al.* Autophagy mediates the mitotic senescence transition. *Genes Dev.* **23**, 798–803 (2009).
16. De Cecco, M. *et al.* Genomes of replicatively senescent cells undergo global epigenetic changes leading to gene silencing and activation of transposable elements. *Aging Cell* **12**, 247–256 (2013).
17. Reeves, R. & Nissen, M. S. The A.T-DNA-binding domain of mammalian high mobility group I chromosomal proteins. A novel peptide motif for recognizing DNA structure. *J. Biol. Chem.* **265**, 8573–8582 (1990).
18. Colombo, D. F., Burger, L., Baubec, T. & Schübeler, D. Binding of high mobility group A proteins to the mammalian genome occurs as a function of AT-content. *PLoS Genet.* **13**, e1007102 (2017).
19. Reeves, R., Leonard, W. J. & Nissen, M. S. Binding of HMG-I(Y) imparts architectural specificity to a positioned nucleosome on the promoter of the human interleukin-2 receptor alpha gene. *Molecular and Cellular Biology* **20**, 4666–4679 (2000).
20. Kuilman, T. & Peeper, D. S. Senescence-messaging secretome: SMS-ing cellular stress. *Nat Rev Cancer* **9**, 81–94 (2009).
21. Bray, S. J. Notch signalling: a simple pathway becomes complex. *Nature Reviews Molecular Cell Biology* **7**, 678–689 (2006).

a

b

Core enrichment
(89 genes downregulated in both NIS and old cells)

Rank	Geneset (GO Biological Process 2017b)	Adj p-val	Z-score	Combined
1	mitotic metaphase plate congression (GO:0007080)	7.315e-18	-2.89	135.32
2	mitotic spindle elongation (GO:0000022)	1.958e-17	-2.77	124.86
3	mitotic sister chromatid segregation (GO:0000070)	1.069e-11	-3.01	93.51
4	mitotic chromosome movement towards spindle pole (GO:0007079)	1.069e-11	-2.96	91.78
5	mitotic chromosome condensation (GO:0007076)	8.401e-12	-2.73	86.72
6	mitotic sister chromatid cohesion (GO:0007064)	6.077e-11	-2.98	86.58
7	mitotic chromosome decondensation (GO:0007083)	2.957e-10	-3.11	84.98
8	mitotic sister chromatid separation (GO:0051306)	5.927e-10	-2.90	77.02
9	attachment of mitotic spindle microtubules to kinetochore (GO:0051315)	1.658e-9	-2.76	70.00
10	mitotic metaphase chromosome recapture (GO:1990942)	2.688e-8	-2.79	62.27

Reviewers Figure R1. Transcriptional signatures in NIS do not reflect the changes observed in ‘old’ cells. **(a)** Gene set enrichment analysis (GSEA) analysis using indicated genesets from MSigDB (top) and the top 100 upregulated and downregulated genes in old cells relative to young cells (bottom, RNA-seq data downloaded from GEO: GSE63577). These gene lists were used for GSEA in RNA-seq data generated from growing and NIS cells. **(c)** Gene ontology analysis (GO biological process 2017b) of genes downregulated in both ‘old’ and NIS cells.

Reviewers Fig. R2

Reviewers Figure R2. Chromatin accessibility reflects gene transcription in RIS and NIS cells. **(a-c)** Overlaps between genes with increased accessibility in RIS and NIS cells versus growing cells and genes that become significantly upregulated by mRNA-seq (adjusted p-value <0.05) in RIS **(a)** and NIS **(b)** versus growing cells. **(c)** Overlap between genes that are both upregulated and more accessible and in RIS cells relative to growing cells with genes that are more accessible and present in the GO term 'inflammatory response' reported in Fig. 4g.

Reviewers Fig. R3

Reviewers Figure R3. Enhancers identified in growing or RIS cells overlap well with enhancers in publicly available annotations. Enhancers for annotation of accessible regions were identified in the current study as regions enriched for both H3K27Ac and H3K4me1 in growing and RIS cells (blue). The number of enhancers in our study that overlap with regions annotated as an enhancer in two public repositories is presented (grey bars). The percentage indicates the percentage of our enhancers that are also present in each repository.

Reviewers Fig. R4

Reviewers Figure R4. DAPT treatment boosts the basal level of SAHF positive cells in a 'growing' population. **(a)** SAHF counts of IMR90 ER:RASG12V cells \pm 10 μ M DAPT for 6 days (in the absence of 4OHT). Statistical significance calculated using a two-sample t-test. $n=3$. **(b)** Representative images of the cells described in (a) stained with DAPI. 3 images per condition (\pm DAPT) are provided. Insets are magnified images of the indicated SAHF positive nuclei (arrows). Scale bar = 50 μ m

Reviewers Fig. R5

Reviewers Figure R5. A combination of THOR and EdgeR was used to detect differentially accessible regions genome-wide. **(a)** Correlation between replicate ATAC-seq samples generated from growing cells. Numbers denote the percentage of peaks in the replicate labelled on the x-axis that overlap with peaks detected in the replicate labelled on the y-axis. **(b)** Genome browser image showing regions detected as significantly more accessible in RIS and/or NIS cells relative to growing cells by EdgeR, THOR or both. Regions detected by only one method are generally less convincing and were excluded from downstream analysis. **(c)** A volcano plot showing regions that alter by EdgeR in the comparison 'N+RIS versus RIS', coloured based on the \log_2 fold change at the same loci enumerated using THOR.

Reviewers Fig. R6

Reviewers Figure R6. HMGA1 motif enrichment analysis around regions that become more accessible in RIS and NIS relative to growing cells. **(a)** The HMGA1 motif identified. **(b)** Unbiased motif enrichment analysis was performed and the motif was found to be significantly enriched in the regions flanking both RIS-driven accessible regions (opened in RIS) and NIS-driven accessible regions (opened in NIS), though the enrichment was many times greater within RIS-driven accessible regions. Windows around the peak centre (± 300 bp), downstream (+500bp) and upstream (-500bp) of the peak centre were analysed.

REVIEWERS' COMMENTS:

Reviewer #1 (Remarks to the Author):

I want to commend the authors for their thoughtful and informative answers to all referees' questions. I am satisfied with the answers and recommend publication as is.

Reviewer #2 (Remarks to the Author):

The authors have taken considerable care in responding to all of the Reviewers's comments, and have provided a detailed and considered response in their rebuttal document. Following the revisions, the originally strong manuscript has been further improved, and the generation of genome edited JAG1 knockout clones adds a further dimension to the work. I believe that this work would be of considerable interest to the readership of Nature Communications, and would be delighted recommend acceptance of this revised manuscript following very minor proof reading changes.

Minor changes

Affiliations – UK. or UK

Fig. or Fig

Ensure that each gene name used within the main body of the text is provided in full e.g. HMGA1
Consistency with regards histone modification format – e.g. H3K27Ac versus H3K4me1 (i.e. should this be H3K3Me1).

Provide full name for LINEs, LTRs, SINEs.

R values: R= or r=

P values: $p < 0.01$ or $P < 0.01$ or $P < 0.01$

< or >: with our without a space after

Concentrations: 100nM or 100 nM (i.e. with or without a space)

Check numbers >999 have a comma e.g. page 24 line 9: 1,758-2,556

Ensure that all sequences have 5' and 3' indicated.

n=3 or n= 3

Page 27 line 21: change 75-bp to 75bp

Page 28 line 7: change 50-bp to 50bp

Reviewer #3 (Remarks to the Author):

The authors addressed all my concerns. They especially made a considerable effort in increasing robustness of their data analyses, and in providing all the relevant methodological details. The revised version of the analyses now fully support authors' claims. I think that in this revised version, the manuscript is suitable for publication in Nature Communication.

We thank all of the reviewers for their time and for their positive feedback. Reviewer #1 and Reviewer #3 have no further questions.

Reviewer #2 (Remarks to the Author):

Minor changes

Affiliations – UK. or UK

Fig. or Fig

Corrected.

Ensure that each gene name used within the main body of the text is provided in full e.g. HMGA1.

When referring to HMGA1 alone, the full name is used. When both HMGA1 and HMGA2 is referred to, we have altered the text to make this clear. i.e. 'HMGA proteins' or 'HMGA genes'.

Consistency with regards histone modification format – e.g. H3K27Ac versus H3K4me1 (i.e. should this be H3K3Me1)

We have changed H3K27Ac to H3K27ac as this is how it most commonly appears.

Provide full name for LINEs, LTRs, SINEs.

Full names have now been provided.

R values: R= or r=

r= has been replaced with R = (with a space)

P values: p <0.01 or P <0.01 or P < 0.01, with or without a space after < (or >)

These have been unified as e.g. p < 0.01 (small 'p' with spaces)

Concentrations: 100nM or 100 nM (i.e. with or without a space)

Concentrations are now quoted as e.g. 100 nM (with spaces)

Check numbers >999 have a comma e.g. page 24 line 9: 1,758-2,556

Commas have been introduced.

Ensure that all sequences have 5' and 3' indicated.

5' and 3' have been added where missing.

n=3 or n= 3

These have been changed so that all are e.g. n = 3 (with spaces)

Page 27 line 21: change 75-bp to 75bp

Page 28 line 7: change 50-bp to 50bp

These have been changed.